# On the "Induction Bias" in Sequence Models

**M.Reza Ebrahimi** [1]  **Michaël Defferrard** [1]  **Sunny Panchal** [1]  **Roland Memisevic** [1]

## Abstract

Despite the remarkable practical success of transformer-based language models, recent work has raised concerns about their ability to perform state tracking. In particular, a growing body of literature has shown this limitation primarily through failures in out-of-distribution (OOD) generalization, such as length extrapolation. In this work, we shift attention to the in-distribution implications of these limitations. We conduct a large-scale experimental study of the data efficiency of transformers and recurrent neural networks (RNNs) across multiple supervision regimes. We find that the amount of training data required by transformers grows much more rapidly with state-space size and sequence length than for RNNs. Furthermore, we analyze the extent to which learned state-tracking mechanisms are shared across different sequence lengths. We show that transformers exhibit negligible or even detrimental weight sharing across lengths, indicating that they learn length-specific solutions in isolation. In contrast, recurrent models exhibit effective amortized learning by sharing weights across lengths, allowing data from one sequence length to improve performance on others. Together, these results demonstrate that state tracking remains a fundamental challenge for transformers, even when training and evaluation distributions match.

## 1. Introduction

State tracking is a key capability of most intelligent systems and of most models of computation. It is the process of monitoring and updating the status of an entity or process with which the system interacts over a period of time. State tracking is particularly important in multi-hop, interactive tasks, such as that of an agent interacting with an interface or of a dialogue system interacting with a user across multiple turns.

State tracking has become a popular area of investigation in recent years, especially in the study of LLM capabilities and failure modes. In this context, numerous studies have shown that transformer-based models are fundamentally limited in their ability to perform state tracking (for example, (Anil et al., 2022; Dziri et al., 2024)). This contrasts with recurrent networks, which excel at state tracking (although their widespread applicability is unfortunately hampered by their relative training inefficiency). The limitations of transformers have been demonstrated as limitations in out-of-distribution (OOD) generalization, specifically length-generalization: after training models on tasks encoded in sequences of a given range of lengths, they were evaluated on sequences with lengths that were not seen during training. In these scenarios, the trained models fail to consistently generate correct outputs on the evaluation data, although they are able to solve the tasks for even unseen sequences in the training range of lengths.

It could be argued that in any real-world use cases, OOD state tracking failures may not be an issue as long as enough training data with step-by-step sequential supervision is available. If training data covers all sequence lengths that may be encountered at inference time, inference can rely entirely on in-distribution generalization. Unfortunately, although this argument is true in principle, it is hard to quantify "enough" in this context. It is also hard to quantify how the amount of training data required for any given task may depend on the length of the sequences and the size of the state space.

To shed light on these questions, in this work we perform a detailed and systematic empirical study of the *in-distribution* performance of transformer-based models and contrast these with recurrent models. To this end, we train and evaluate a range of representative models on a range of simple state-tracking tasks. Independently varying sequence length and size of the state space in these tasks then allows us to discover regularities in the dependence of generalization error on these parameters. This in turn makes it possible to obtain a crude "lower bound" on the minimal amount of training data likely required to solve such tasks.

A key difference between transformer-based and recurrent

---

[1]Qualcomm AI Research is an initiative of Qualcomm Technologies, Inc. Correspondence to: Reza Ebrahimi <ebrahimi@qualcomm.com>.

models is that, at every time step, the former compute outputs by applying a function that depends on *all* inputs and outputs generated previously (the context window), making it possible, in principle, to recalculate the required state from the past information globally at each time step. Recurrent networks, on the other hand, compute outputs by applying a function that depends on only the current hidden state, making it impossible to perform such a re-calculation. This makes it strictly necessary for a recurrent network to encode any relevant information from the past within a single hidden state vector. This inductive bias encourages a recurrent network to incorporate the information from the current time step into its representation of state at the moment where this information is available. Conversely, it discourages it from "saving" this information off to determine future state updates globally from the past information. The immediate state updates thus encourage the network to process the input sequence step-by-step, making any state update explicit as soon as this is possible, rather than potentially deferring such updates to a later point in time.

Such step-by-step state updates are a natural inductive bias (Mitchell, 1997) in the context of simple state-tracking tasks, as they make it possible to reduce complex multi-step dependencies to a sequence of single-step computations. They also allow a model to share weights across multiple different sequence lengths, as it breaks state updates into single-step, repeatable computations. By analogy to the induction step in a mathematical proof, we shall refer to this kind of inductive bias in this work as "induction bias" (sic).

Formally, the presence of the induction bias in a model means that the joint distribution over tokens, conditioned on the most recent hidden state, factorizes, such that $p(x_{t+1}|x_1, \ldots, x_t, h_t) = p(x_{t+1}|h_t)$, where $x_t$ is the $t$-th token and $h_t$ the hidden state in time step $t$, representing a *minimal sufficient statistic* for determining $x_{t+1}$.

We show that the presence (or respectively absence) of an induction bias, or its relative strength, provides a simple explanation for a wide range of the empirical findings we present.

Key take-aways from our study include the following:

- We show that there is a distinct difference between the supervision regimes in which transformers and recurrent networks perform well in-distribution.

- We show that transformers can relatively efficiently learn state-tracking tasks in-distribution on one (fixed) sequence length at a time, but generalizing in-distribution over multiple sequence lengths requires significantly more training data.

- We present evidence that, unlike recurrent networks, transformers tend to fail at sharing parameters across sequence lengths and instead learn separate solution mechanisms for different lengths.

- We show that the degree of knowledge transfer across multiple different sequence lengths in the in-distribution setting is highly correlated with the ability of a model to length-generalize.

### 1.1. Related Work

A range of studies has shown that transformer-based sequence models fail to length-generalize in state-tracking tasks (Anil et al., 2022; Deletang et al., 2023; Dziri et al., 2024; Abbe et al., 2024; Ebrahimi et al., 2024). Unlike our work, these studies solely discuss OOD scenarios, while we discuss in-distribution data efficiency instead. Closely related, Chang & Bisk (2025) study inductive counting, where success requires learning a reusable increment operator rather than memorizing seen cardinalities. Their finding that traditional RNNs generalize naturally, while transformers and SSMs struggle to generalize inductively, aligns with our view that recurrent architectures provide an induction bias toward reusable state updates.

The inability to length-generalize in state-tracking tasks has also been shown to hold for most existing state-space models (SSM) (Sarrof et al., 2024; Merrill et al., 2024; Cirone et al., 2024; Shakerinava et al., 2026). However, recent work has shown that making the hidden-to-hidden transition matrix in the SSM input-dependent and non-diagonal can recover the ability to length-generalize (Fan et al., 2024; Grazzi et al., 2025; Ebrahimi & Memisevic, 2025; Terzić et al., 2025a;b).

Liu et al. (2023); Li et al. (2025) show that transformers solve state-tracking tasks in-distribution by making use of parallel mechanisms reminiscent of associative scan. While this view can help explain the OOD failures of these models, it also hints at the absence of an "induction" bias which affects data efficiency as we show in this work.

Most closely related to our task formulation, Marchetti et al. (2026) study sequential group composition and show that shallow feed-forward networks require width exponential in sequence length, whereas recurrent models can exploit associativity to compose intermediate states with width independent of sequence length. Our work is complementary: we use group-composition tasks to quantify in-distribution sample complexity and cross-length weight sharing across transformer and recurrent architectures.

## 2. Methodology

We formalize state tracking as the problem of maintaining a latent state under a sequence of observed updates. In group-theoretic terms, given a group $(G, \circ)$ and a sequence of updates $g_1, \ldots, g_T \in G$, the task is to compute the cumulative product $g_1 \circ g_2 \circ \cdots \circ g_T \in G$.

This composition problem abstracts the essence of state tracking: each update to a world state, such as a chess move, a variable assignment, or an entity swap, can be modeled as an element of an algebraic structure, and applying updates sequentially corresponds to multiplying those elements. For example, parity corresponds to a two-state system whose state is flipped when observing input 1.

In this work, we instantiate this framework with two canonical cases: modular addition over $\mathbb{Z}_m$ as the commutative setting, and permutation composition over the symmetric group $S_m$ as a non-commutative counterpart. The latter is motivated by prior work showing that permutation composition over $S_5$ can be reduced to chess state tracking and captures structure also present in code evaluation and entity tracking (Merrill et al., 2024).

**Tasks:** We consider the task of modular addition, where a model is provided a sequence of $n$ integers $\mathbf{x} = (x_1, x_2, \ldots, x_n)$ with each $x_i$ drawn uniformly at random from $\mathbb{Z}_m = \{0, 1, \ldots, m-1\}$. The objective is to compute the sum of the sequence modulo $m$:

$$y = \left( \sum_{i=1}^{n} x_i \right) (\mathrm{mod}\ m), \quad x_i \in \mathbb{Z}_m.$$

For $m = 2$, the task reduces to computing the parity of a binary sequence. From an algebraic perspective, modular addition over $\mathbb{Z}_m$ (cyclic group) serves as the canonical representative for commutative operations, as every finite abelian group is isomorphic to a direct product of such cyclic groups.

We also experiment with non-commutative operations by considering the task of permutation composition over the symmetric group $S_5$. This task serves as the canonical non-commutative counterpart for state tracking, as by Cayley's Theorem, every finite group is isomorphic to a subgroup of a symmetric group (Dummit & Foote, 2003). For additional details and empirical results on the permutation composition task, we refer the reader to Appendix Section B.2.

These synthetic group tasks allow us to isolate the state-tracking computation while independently controlling the state-space size and sequence length, without confounding factors from language understanding, dataset artifacts, or memorization.

**Length Distributions:** For each generated sample, we first determine the sequence length $n \in \{2, \ldots, L\}$, where $L$ denotes the maximum sequence length. We then sample a sequence $\mathbf{x} \in \mathbb{Z}_m^n$ *without replacement* to ensure that every sample in the dataset is unique. We use three distinct strategies for length selection:

1. *Fixed:* The length is held constant at $n = L$.

2. *Uniform:* Lengths are sampled uniformly at random

from the set $\{2, \ldots, L\}$.

3. *Short-to-Long:* Sequences are sampled in ascending order of length, exhausting the available sequences for length $n$ before proceeding to $n + 1$.

**Task Formats:** We consider three task formats that vary in the density and structure of the supervision signal. Let $s_k = (\sum_{i=1}^{k} x_i) (\mathrm{mod}\ m)$ denote the $k$-th partial sum of the input sequence. The formats, illustrated in Figure 1, are defined as follows:

1. *Outcome Supervision:* The model is provided the input sequence $\mathbf{x}$ and is trained to predict only the final sum $s_n$. This format provides no intermediate supervision, requiring the model to discover the latent computational logic of the task on its own during training.

2. *Chain-of-Thought (CoT):* The model is trained to generate the sequence of intermediate partial sums $(s_1, s_2, \ldots, s_n)$ following the input sequence. This decomposes the task into a sequence of iterative applications of the operator.

3. *Aligned Chain-of-Thought (ACoT):* The model is tasked to output, for each input token $x_i$, the corresponding partial sum $s_i$. While conceptually similar to the scratchpad, this format provides per-token supervision that is aligned with the input. This format is similarly used in prior work (Merrill et al., 2024; Li et al., 2025; Zhang et al., 2025) and is also referred to as *state-supervision* or *token-tagging*.

Unlike outcome supervision, both CoT and ACoT constitute a form of *process supervision*, as they provide explicit training signals for the intermediate solution steps.

**Sample Efficiency:** To quantify the data efficiency of a model under a specific task configuration, we define the *minimal sample size* $N^*$ required to learn the task reliably. Let $\mathcal{D}_N$ denote a training set of cardinality $N$, and let $\mathcal{L}_{\mathrm{val}}(\phi; \mathcal{D}_N)$ denote the validation loss of a model trained on $\mathcal{D}_N$ using hyperparameter configuration $\phi \in \Phi$.

We consider a task successfully learned if the minimum validation loss over the hyperparameter grid falls below a convergence threshold:

$$\min_{\phi \in \Phi} \mathcal{L}_{\mathrm{val}}(\phi; \mathcal{D}_N) \leq \epsilon, \tag{1}$$

where $\Phi$ is a predefined grid of learning rates and random seeds, and $\epsilon$ is a convergence threshold. Formally, we define $N^*$ as the smallest training set size satisfying this criterion:

$$N^* = \min \left\{ N \in \mathbb{N} : \min_{\phi \in \Phi} \mathcal{L}_{\mathrm{val}}(\phi; \mathcal{D}_N) \leq \epsilon \right\}. \tag{2}$$

In practice, we estimate $N^*$ by performing a binary search over the training set size. Note that both training and valida-

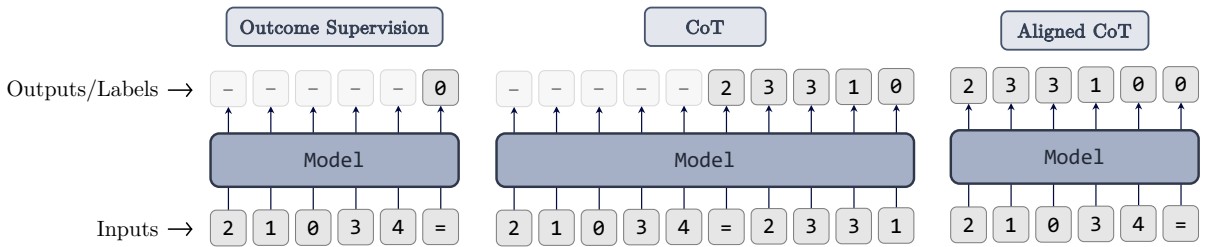

*Figure 1.* Example of the three task formats for the addition modulo 5 task applied to the sequence 2 1 0 3 4.

tion samples are drawn from the same underlying data generation process, and therefore $N^*$ reflects the *in-distribution* sample efficiency. In addition to low validation loss, we also consider perfect validation accuracy as an alternative success criterion and find no meaningful difference in the results. For the results in Section 4, we use the latter.

**Models:** We compare multi-layer decoder-only transformer architecture (Vaswani et al., 2017), with recurrent alternatives: Long Short-Term Memory (LSTM) (Hochreiter & Schmidhuber, 1997) and dense state-space models (Dense-SSMs) (Fan et al., 2024; Terzić et al., 2025a; Ebrahimi & Memisevic, 2025). For the weight-sharing analysis in Section 4, we additionally include Mamba (Gu & Dao, 2024) as a representative selective state-space model.

In Dense-SSMs, the state transition matrix is dense and fully input-dependent, a property shown to support effective state tracking in linear recurrent models (Merrill et al., 2024). We adopt the variant used by Ebrahimi & Memisevic (2025), in which input–state interactions are purely multiplicative with no additive terms:

$$h_t = A_{x_t} h_{t-1}, \tag{3}$$

where the transition matrix $A_{x_t}$ is given by a linear function of the input $x_t$. This architecture is commonly referred to as a *bilinear RNN*, since $h_t$ depends bilinearly on the input and the previous hidden state.

**Experimental Setup:** We perform a large-scale systematic evaluation of data efficiency on synthetic state-tracking tasks. To estimate $N^*$, we use a hybrid binary–geometric search procedure (see Algorithm 1) that evaluates candidate sample sizes over at most 20 steps, training models across a hyperparameter grid consisting of 3 learning rates and 5 random seeds (15 configurations total for each size $N$). A sample size is considered successful if at least one configuration achieves validation loss below $\epsilon = 10^{-4}$. Each model is trained for at most 250k optimization steps, independent of the training set size $N$, with early stopping once the validation-loss success criterion is met. This amounts to over *200,000 training runs* for the results reported in this paper, excluding development runs.

The transformer model used is based on the GPT-2 archi-

tecture (Radford et al., 2019) with 6 layers and a model (embedding/hidden) dimension of 256. Both the LSTM and Dense-SSM models use a single-layer recurrent cell followed by a linear classification head. We use input and hidden dimension of 768 for LSTM and 256 for the Dense-SSM. The Mamba model has 6 layers and model dimension 256. Additionally, we experiment with a 2-layer transformer and LSTM with hidden dimension of 256, with results provided in the Appendix Section B.3.

We ensure that the training and validation sets are strictly disjoint. The validation set contains 2,000 samples (or at most 20% of the available data) and remains identical across different training set sizes, except for variations introduced by the random seed. In addition, we always use at most 20% of the available samples at each sequence length for validation, with the remainder reserved exclusively for training. Also, for all tasks, multi-digit integers are represented as single tokens during tokenization. Additional implementation details are provided in Appendix A.

## 3. In-Distribution Data Efficiency

We perform the above binary search procedure to identify the minimal dataset size ($N^*$) across all combinations of maximum sequence length $L \in \{5, 10, 20, 30\}$ and modulus $m \in \{2, 3, 5, 10, 15, 20, 50, 75, 100\}$, for each of the three task formats, length distributions, and models described earlier. The results are summarized in Table 1. For ease of comparison, we also visualize selected slices of this table in figures throughout this section. The fitted curves in these figures are intended only as visual guides. Comparable results for models of different sizes are reported in Appendix B.3. From the table we can infer the following key observations:

> **Observation 3.1**
>
> Transformers prefer non-aligned supervision (Chain of Thought).

We observe a clear preference of transformers for CoT over the Aligned CoT format. For example, at $m = 5$ and $L = 20$, CoT requires 1.7K samples, while Aligned CoT requires 2M, an order-of-magnitude increase in sample complexity.

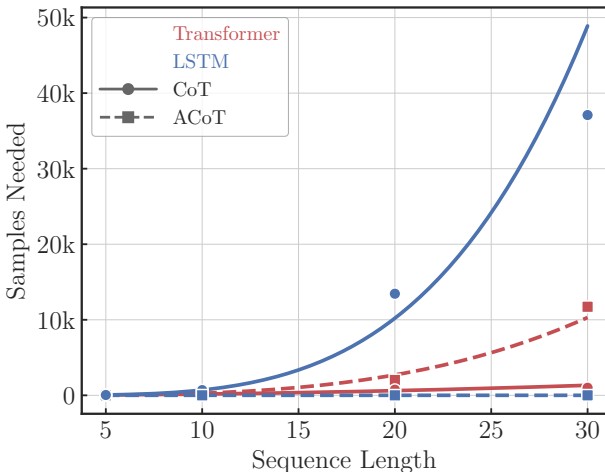

*Figure 2.* Minimal dataset size for the uniform length distribution with $m = 2$ (parity). RNNs favor ACoT, whereas transformers favor CoT.

Figure 2 further illustrates this gap in sample requirements for the case $m = 2$ (parity) across the two formats.

It has been hypothesized that by outputting intermediate steps autoregressively, the model can attend to its own previous outputs, effectively simulating a larger depth circuit (Li et al., 2024), and the results confirm this hypothesis. In contrast, Aligned Chain-of-Thought forces the model to compress the computation into a single forward pass per token without the benefit of re-attending to intermediate results, which appears less aligned with the transformer's non-recurrent nature.

> **Observation 3.2**
>
> Recurrent models prefer aligned supervision (Aligned Chain-of-Thought).

Conversely, recurrent models (LSTMs and Dense-SSMs) demonstrate superior sample efficiency when trained with the Aligned CoT (ACoT) format, which provides supervision aligned with the evolution of the hidden state (see also Figure 2).

In contrast, RNNs struggle with CoT, which is likely due to their recall bottleneck (Wen et al., 2025; Phan et al., 2025): a model must output the sequence of partial sums $(s_1, \ldots, s_n)$ *after* processing the entire input sequence. This effectively requires it to unroll the chain of intermediate computations from the beginning. In fact, we note that under the CoT format, recurrent models even fail to generalize to longer sequences despite their sequential inductive bias (see Table 2 for length-generalization results). The task thereby becomes bottlenecked by the model's limited memory capacity rather than its state-tracking ability.

> **Observation 3.3**
>
> Recurrent models outperform transformers in the absence of intermediate supervision.

In the Outcome Supervision setting, the model must implicitly infer the latent algebraic structure of the task solely from the final solution, without any guidance on the intermediate steps. This requires the model to effectively marginalize over unobserved computational paths with difficulty scaling with both the state space size $m$ and sequence length $n$.

We observe that recurrent models significantly outperform transformers in this regime. While transformers fail to converge for all but the most trivial configurations (very small $m$ and $n$), the recurrent architectures successfully learn the task for higher moduli and extended sequence lengths, achieving convergence with orders of magnitude fewer training samples. Figure 3 illustrates this behavior for the parity case ($m = 2$).

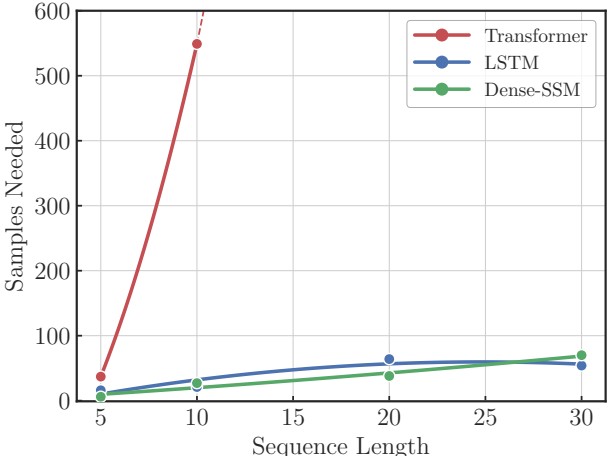

*Figure 3.* $N^*$ for the outcome supervision format with a uniform length distribution and $m = 2$ (parity). In the absence of intermediate supervision, single-layer RNNs significantly outperform the 6-layer transformer.

> **Observation 3.4**
>
> With intermediate supervision, longer sequences improve the data efficiency of recurrent models but not transformers.

Intuitively, under formats with intermediate supervision (CoT or ACoT), longer sequences should improve sample efficiency. This is because with intermediate solutions, the effective amount of supervised tokens increases linearly with sequence length.

We validate this hypothesis in recurrent models trained with Aligned Chain-of-Thought: the fixed length distribution (comprising only longest sequences) yields the highest data efficiency, followed by uniform, and finally short-to-long.

*Table 1.* Minimal number of training samples required to learn the modulo addition task. A dash (–) indicates that the task was not learned at the maximum training set size.

| Model | Format | Length Modulus | Fixed 5 | 10 | 20 | 30 | Uniform 5 | 10 | 20 | 30 | Short-to-Long 5 | 10 | 20 | 30 |
|---|---|---|---|---|---|---|---|---|---|---|---|---|---|---|
| Transformer | Outcome Supervision | 2 | 19 | 364 | – | – | 37 | 549 | – | – | 45 | 913 | – | – |
| | | 3 | 119 | – | – | – | – | – | – | – | 243 | – | – | – |
| | | 5 | 1.1K | – | – | – | 2.6K | – | – | – | 1.5K | – | – | – |
| | | 10 | – | – | – | – | – | – | – | – | – | – | – | – |
| | | 15 | – | – | – | – | – | – | – | – | – | – | – | – |
| | | 20 | – | – | – | – | – | – | – | – | – | – | – | – |
| | | 50 | – | – | – | – | – | – | – | – | – | – | – | – |
| | | 75 | – | – | – | – | – | – | – | – | – | – | – | – |
| | | 100 | – | – | – | – | – | – | – | – | – | – | – | – |
| | Chain-of-Thought | 2 | 10 | 16 | 19 | 20 | 36 | 198 | 744 | 1K | 39 | 824 | 1M | – |
| | | 3 | 16 | 18 | 21 | 25 | 78 | 465 | 1.2K | 1.4K | 108 | 28.8K | – | – |
| | | 5 | 33 | 30 | 31 | 34 | 148 | 1.1K | 1.7K | 1.7K | 647 | 2.4M | – | – |
| | | 10 | 94 | 66 | 62 | 70 | 427 | 2.3K | 3.4K | 4K | 11K | – | – | – |
| | | 15 | 166 | 116 | 78 | 107 | 709 | 2.5K | 5K | 5.1K | 54.3K | – | – | – |
| | | 20 | 377 | 178 | 116 | 135 | 1K | 3.8K | 5.6K | 7.4K | 168.6K | – | – | – |
| | | 50 | 1.1K | 678 | 553 | 470 | 2.6K | 8.6K | 12K | 11.2K | 6.4M | – | – | – |
| | | 75 | 1.7K | 1.2K | 1K | 946 | 4.2K | 11K | 18.1K | 16.1K | – | – | – | – |
| | | 100 | 2.5K | 1.9K | 1.6K | 1.6K | 5.8K | 13.2K | 21.1K | 19.1K | – | – | – | – |
| | Aligned Chain-of-Thought | 2 | 16 | 121 | 644 | 1.2K | 27 | 313 | 2K | 11.7K | 19 | 856 | 1M | – |
| | | 3 | 91 | 2K | 3.7K | 4K | 180 | 8.3K | 66.8K | – | 207 | 29.7K | – | – |
| | | 5 | 528 | 3.9K | 9.2K | 20.8K | 671 | 14.5K | 2M | – | 909 | 2.4M | – | – |
| | | 10 | 1.3K | 6K | 15.5K | 131.4K | 1.7K | 32.3K | – | – | 11.2K | – | – | – |
| | | 15 | 1.5K | 7.9K | 23.3K | – | 2.4K | 61.5K | – | – | 55K | – | – | – |
| | | 20 | 2.6K | 21.9K | 85.3K | – | 4.5K | 190.9K | – | – | 172.6K | – | – | – |
| | | 50 | 13K | 1M | – | – | 22K | – | – | – | 6.7M | – | – | – |
| | | 75 | 21.1K | 15.2M | – | – | 102.1K | – | – | – | – | – | – | – |
| | | 100 | 23.6K | – | – | – | 8M | – | – | – | – | – | – | – |
| LSTM | Outcome Supervision | 2 | 14 | 101 | 252 | 309 | 16 | 21 | 64 | 54 | 12 | 9 | 13 | 16 |
| | | 3 | 195 | 620 | 1.7K | – | 78 | 122 | 317 | 526 | 83 | 90 | 90 | 100 |
| | | 5 | 2K | 5.9K | – | – | 387 | 707 | – | – | 381 | 22.7K | – | – |
| | | 10 | 6.4K | – | – | – | 2.2K | 8.6K | – | – | 2.2K | – | – | – |
| | | 15 | 18.8K | – | – | – | 11.8K | – | – | – | – | – | – | – |
| | | 20 | – | – | – | – | – | – | – | – | – | – | – | – |
| | | 50 | – | – | – | – | – | – | – | – | – | – | – | – |
| | | 75 | – | – | – | – | – | – | – | – | – | – | – | – |
| | | 100 | – | – | – | – | – | – | – | – | – | – | – | – |
| | Chain-of-Thought | 2 | 20 | 307 | 4.7K | 8.6K | 45 | 692 | 13.4K | 37.1K | 48 | 843 | 1.1M | – |
| | | 3 | – | 2.9K | 9.3K | 14K | – | 7.4K | 40.3K | 5.2M | – | 29K | – | – |
| | | 5 | 1.6K | 10.3K | 20.4K | 44K | 2K | 28.4K | 68.1K | 2.1M | 2.1K | – | – | – |
| | | 10 | 6.1K | 15.7K | 29.1K | – | 14.3K | 544.4K | 500K | – | 16.8K | – | – | – |
| | | 15 | 15.8K | 21.1K | 42K | – | 32K | 67K | – | – | 65.6K | – | – | – |
| | | 20 | 21.2K | 29.6K | 54.3K | – | 40.7K | 409.1K | 14M | – | 215.4K | – | – | – |
| | | 50 | 43K | 50.4K | – | – | 206K | 2.2M | – | – | 10.4M | – | – | – |
| | | 75 | 56.5K | 58K | – | – | 4M | – | – | – | – | – | – | – |
| | | 100 | 84.2K | 9.5M | – | – | – | – | – | – | – | – | – | – |
| | Aligned Chain-of-Thought | 2 | 4 | 6 | 2 | 2 | 4 | 4 | 2 | 2 | 9 | 7 | 7 | 8 |
| | | 3 | 10 | 8 | 8 | 6 | 18 | 12 | 6 | 8 | 27 | 28 | 34 | 34 |
| | | 5 | 95 | 70 | 39 | 31 | 153 | 107 | 80 | 47 | 142 | 148 | 253 | 235 |
| | | 10 | 278 | 194 | 146 | 130 | 447 | 313 | 247 | 232 | 685 | 1.3K | 1.8K | 2K |
| | | 15 | 534 | 478 | 409 | 371 | 865 | 631 | 567 | 500 | 1.5K | 3.9K | 5.2K | 7.1K |
| | | 20 | 798 | 672 | 652 | 798 | 1.3K | 934 | 870 | 820 | 2.7K | 9.4K | 13.4K | 25.9K |
| | | 50 | 3.7K | 4.6K | 6K | 7.4K | 4.1K | 5.3K | 6.1K | 6.9K | 14.7K | 156.2K | – | – |
| | | 75 | 7.4K | 9.2K | 12.2K | 13.3K | 8.4K | 11.3K | 14.2K | 16.8K | 41.1K | 697.4K | – | – |
| | | 100 | 12.8K | 16.6K | 21.9K | 23.7K | 13.3K | 19.4K | 24.3K | 28.5K | 99.7K | – | – | – |
| Dense-SSM | Outcome Supervision | 2 | 13 | 58 | 33 | 70 | 6 | 27 | 38 | 70 | 9 | 10 | 12 | 12 |
| | | 3 | 56 | 390 | 8K | 2.2K | 31 | 69 | 209 | 265 | 24 | 32 | 41 | 41 |
| | | 5 | 323 | 3.8K | 10.9M | – | 118 | 285 | 1K | 1.1K | 69 | 83 | 132 | 163 |
| | | 10 | 2.7K | 8.3M | – | – | 573 | 2.2K | 4.5K | – | 321 | 452 | 678 | 1.1K |
| | | 15 | 6.5K | – | – | – | 1.6K | – | 12.1K | – | 970 | 1.4K | 3.7K | 3.8K |
| | | 20 | 15.7K | – | – | – | 3K | – | – | – | 2K | 8.4K | 8.8K | 9K |
| | | 50 | – | – | – | – | 15.6K | – | – | – | 11K | – | – | – |
| | | 75 | – | – | – | – | – | – | – | – | – | – | – | – |
| | | 100 | – | – | – | – | – | – | – | – | – | – | – | – |
| | Chain-of-Thought | 2 | – | 172 | 2.1K | – | – | 511 | – | – | – | 875 | – | – |
| | | 3 | 110 | 1K | – | – | 216 | 6.2K | – | – | 234 | 29.8K | – | – |
| | | 5 | 538 | 5.9K | – | – | 1.5K | 19.5K | – | – | 1.5K | 2.4M | – | – |
| | | 10 | 3.4K | – | – | – | 12.5K | – | – | – | 14.8K | – | – | – |
| | | 15 | 12.6K | – | – | – | 21.2K | – | – | – | 56.2K | – | – | – |
| | | 20 | 21.9K | – | – | – | – | – | – | – | 2.5M | – | – | – |
| | | 50 | – | – | – | – | – | – | – | – | – | – | – | – |
| | | 75 | – | – | – | – | – | – | – | – | – | – | – | – |
| | | 100 | – | – | – | – | – | – | – | – | – | – | – | – |
| | Aligned Chain-of-Thought | 2 | 2 | 2 | 3 | 1 | 3 | 2 | 2 | 1 | 6 | 4 | 4 | 4 |
| | | 3 | 4 | 6 | 4 | 4 | 7 | 6 | 4 | 4 | 9 | 11 | 10 | 10 |
| | | 5 | 18 | 9 | 6 | 6 | 24 | 17 | 10 | 10 | 30 | 26 | 25 | 25 |
| | | 10 | 109 | 41 | 21 | 17 | 148 | 74 | 56 | 27 | 102 | 101 | 101 | 101 |
| | | 15 | 219 | 109 | 66 | 37 | 412 | 186 | 101 | 70 | 228 | 225 | 225 | 225 |
| | | 20 | 438 | 233 | 124 | 78 | 633 | 382 | 241 | 116 | 396 | 397 | 405 | 405 |
| | | 50 | 3.2K | 1.7K | 1.1K | 874 | 4.5K | 3.2K | 2K | 1.6K | 2.5K | 2.5K | 2.5K | 2.5K |
| | | 75 | 8.1K | 4K | 3.1K | 2.1K | 10.8K | 6.8K | 5.1K | 3.8K | 5.6K | 5.6K | 5.6K | 5.6K |
| | | 100 | 12.5K | 6K | 4.3K | 3K | 16.1K | 10.2K | 6.7K | 6K | 10K | 10K | 12.8K | 16.4K |

Furthermore, in the uniform setting, we find that recurrent models trained with ACoT require fewer data points as the maximum sequence length $L$ increases, as expected. In contrast, transformers trained with CoT fail to leverage this additional supervision. This trend is also evident in Figure 4, which compares sample complexity on the parity task.

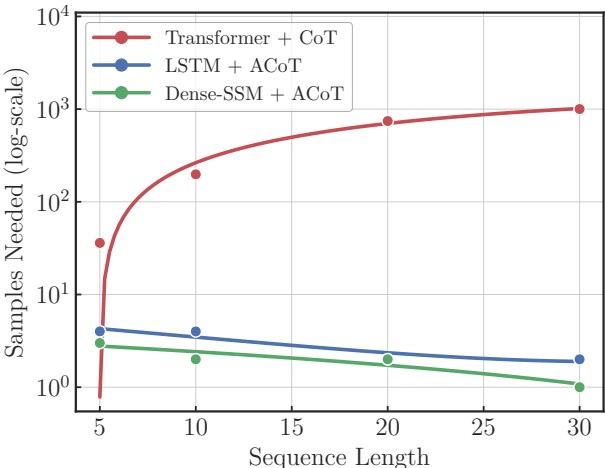

*Figure 4.* Sample complexity (log scale) for transformers trained with CoT and RNNs with ACoT on the parity task. RNNs exhibit the expected improvement in sample efficiency with increasing sequence length, while transformers fail to leverage the additional supervision.

> **Observation 3.5**
>
> With outcome supervision, short sequences are more valuable for learning than long sequences in recurrent models.

In the Outcome Supervision setting, we compare the data

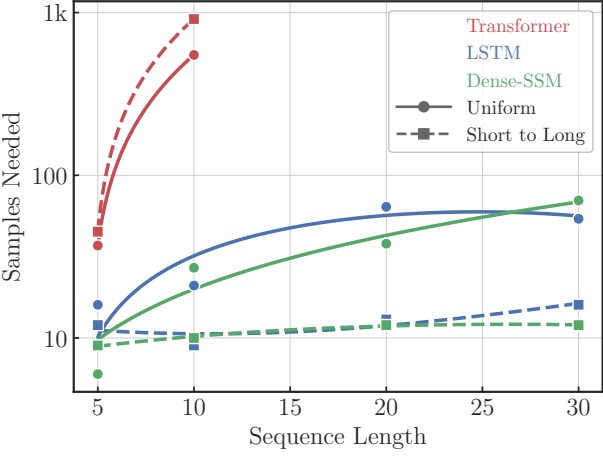

*Figure 5.* Sample complexity (log scale) in the Outcome Supervision format for the uniform and short-to-long length setting, with $m = 2$ (parity). Recurrent models require fewer training samples under the short-to-long setting, indicating that shorter sequences provide a stronger learning signal.

requirements under the uniform and short-to-long length distributions. Recall that these two distributions differ only in the order in which samples are presented during training. We observe that recurrent models require fewer samples in the short-to-long setting, suggesting that shorter sequences provide a stronger learning signal than longer sequences for these models. This effect is illustrated in Figure 5 for the case $m = 2$.

## 4. Weight Sharing Across Sequence Length

A key hypothesis for why recurrent networks dominate transformers with respect to data efficiency, as shown in the previous section, is that their "induction bias" encourages step-by-step updates to their representations of state. This, in turn, should allow the model to share the same solution mechanisms across the whole sequence length.

In this section, we investigate the extent to which the learned mechanisms are shared across different sequence lengths. Specifically, we examine whether the model develops length-specific heuristics, effectively "specialized circuits" for fixed-length sequences, or whether it has internalized the inherent inductive structure of the task. The latter implies the discovery of a transition operator that can be applied iteratively. Assessing the degree of this cross-length sharing is critical for understanding a model's inductive capacity and its ability to generalize to sequence lengths not encountered in the training distribution.

We quantify the cross-length mechanism sharing through the lens of sample efficiency. Intuitively, if a model utilizes a shared mechanism (e.g., a transition operator) across varying lengths, the sample cost to learn the task over a distribution of lengths should be significantly lower than the sum of costs to learn each length individually. This is due to the *amortization* of the learning cost: the data required to learn the operation at length $n$ simultaneously contributes to the model's learning at length $n + k$.

Formally, we compare the total number of training examples required for a model to simultaneously learn the task for all sequence lengths $n \in \{2, \ldots, L\}$ (the joint task, trained on the mixed length distribution) against the sum of samples required by $L - 1$ independent models, each optimized for a single fixed length. Let $N^*_{\text{joint}}$ denote the minimal sample size required for the joint task, and $N^*_n$ denote the minimal sample size for a model trained and evaluated exclusively on sequences of length $n$. We define the *Sharing Factor $\kappa$* as:

$$\kappa = \frac{\sum_{n=2}^{L} N^*_n}{N^*_{\text{joint}}} \quad (4)$$

The value of $\kappa$ provides insight into the extent of across-length mechanism sharing:

- $\kappa > 1$ indicates mechanism sharing and amortized

learning. This suggests the model has internalized the inductive nature of the task, and data from one sequence length accelerates the acquisition of the task across the entire distribution.

- $\kappa \approx 1$ suggests that the model learns length-specific solutions in isolation, effectively partitioning capacity into independent circuits.
- $\kappa < 1$ represents a regime of destructive interference. In this case, the length-specific solutions compete for model capacity, making it more data-efficient to train separate models for each length than to optimize a single model for the joint task.

Figure 6 illustrates the sample complexity to learn addition modulo 5, for all sequence lengths ($N^*_{\text{joint}}$), compared against the cumulative samples required by independent models trained on individual sequence lengths ($\sum_{n=2}^{L} N^*_n$), for $L \in \{2, \ldots, 10\}$. We evaluate these metrics for modular addition with $m = 5$ across the three previously defined task formats, and draw the following key observations. Compara-

ble results for the permutation composition task (symmetric group $S_5$) are reported in Appendix Section B.2. For this analysis, we also include Mamba (Gu & Dao, 2024) as a representative selective state-space model, motivated by prior work showing that common SSM variants can exhibit state-tracking limitations similar to transformers (Merrill et al., 2024). Additional transformer variants based on Llama 3 (Grattafiori et al., 2024) and Pythia (Biderman et al., 2023) are reported in Appendix Section B.4.

> **Observation 4.1**
>
> Transformers have low sharing factor for all task formats.

As demonstrated, we observe a low sharing factor in transformers across all task formats, with $\kappa \approx 1$ or $\kappa < 1$ in all cases. Notably, in the Chain-of-Thought (CoT) setting, despite being the transformer's most efficient task configuration, we observe an extreme case of length isolation ($\kappa = 0.28$).

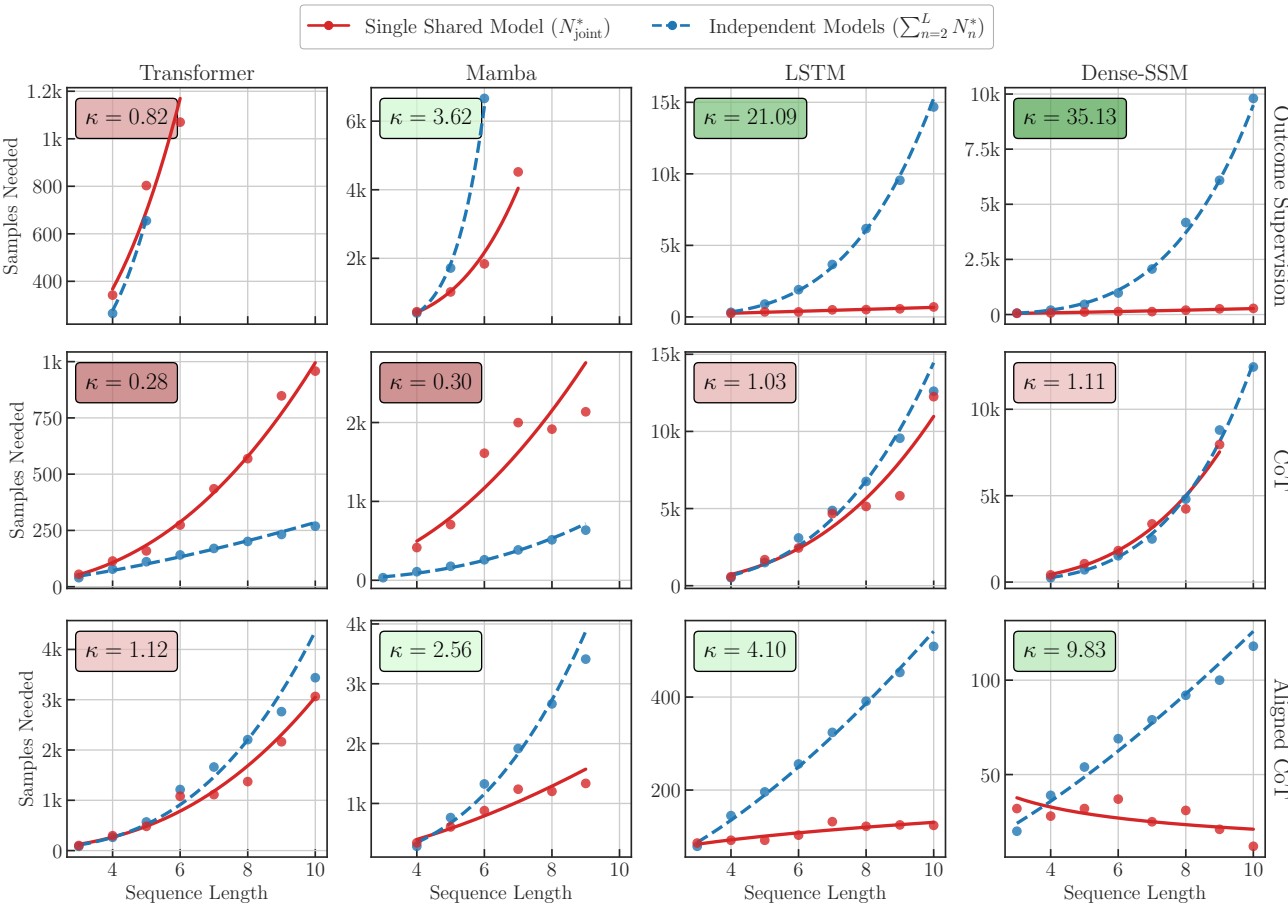

*Figure 6.* Sample complexity comparison between training a single model jointly across all sequence lengths and the cumulative sample complexity of independently trained models for each sequence length, together with the corresponding sharing factor, for the task of addition modulo 5. The results suggest that transformers, and to a lesser extent Mamba, learn largely isolated solutions for each sequence length.

Mamba exhibits slightly stronger sharing than the transformer in the Outcome Supervision and ACoT settings, but remains far below the LSTM and Dense-SSM. Under CoT, it similarly shows poor cross-length sharing.

> **Observation 4.2**
>
> Transformers show destructive interference with CoT.

The observed sharing factor of $\kappa \ll 1$ for transformers and Mamba with CoT indicates a regime of destructive interference where length-specific solutions compete for model capacity, such that training on a diverse length distribution is substantially less data-efficient than training independent models on each length.

> **Observation 4.3**
>
> Recurrent networks have high sharing factors in their preferred task formats.

In contrast, both recurrent models exhibit clear evidence of mechanism sharing and amortized learning across sequence lengths ($\kappa \gg 1$) under the Outcome Supervision and Aligned Chain-of-Thought formats. In the Chain-of-Thought format, this sharing largely disappears ($\kappa \approx 1$), likely due to the previously discussed recall bottleneck. Unlike transformers, however, recurrent models do not exhibit destructive interference in this case.

> **Observation 4.4**
>
> Longer sequences increase data efficiency for Dense-SSMs.

As noted in the previous section, the sample requirement for the Dense-SSM under ACoT *decreases* as the maximum sequence length $L$ increases. This indicates that through cross-length mechanism sharing, the model leverages the higher density of supervision signals in longer sequences.

> **Observation 4.5**
>
> OOD generalization implies high sharing factor, and vice versa.

Interestingly, we observe a consistent correlation between the sharing factor $\kappa$ and length generalization: cases with high sharing factor ($\kappa \gg 1$) correspond to those in which the model learns a length-generalizable solution (see Table 2). Conversely, cases with low sharing factor ($\kappa \leq 1$) are precisely those in which the learned solution fails to extrapolate beyond the training sequence lengths.

This provides additional evidence that in-distribution data efficiency and circuit sharing are fundamental implications of length generalization in state tracking.

## 5. Conclusions

Our study indicates that state tracking poses severe challenges for transformer-based sequence models not only out-of-distribution but also in-distribution: They require extraordinarily large amounts of training data to generalize on simple tasks and require Chain-of-Thought supervision to learn in-distribution on even moderate sequence lengths. This suggests that end-to-end learning in applied "agentic" scenarios, such as robotics or GUI control, could be even more challenging. The fact that data requirements scale with sequence length may also help explain well-known challenges at large context lengths ("context rot").

**Limitations.** We study performance across a limited, albeit representative, set of models and synthetic tasks. These tasks isolate state tracking and allow precise control over state-space size and sequence length, but they do not capture all sources of complexity present in real-world sequential prediction problems. At the same time, the large search space over parameters already requires over *200,000* individual training runs for the current set of models and tasks, limiting the number of architectures and task families we can evaluate directly. Understanding how the data-efficiency and mechanism-sharing patterns observed here transfer to more realistic domains, such as code execution, interaction histories, or embodied control, remains an important direction for future work.

## Impact Statement

This paper presents work whose goal is to advance the field of Machine Learning. There are many potential societal consequences of our work, none which we feel must be specifically highlighted here.

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

# A. Implementation Details

## A.1. Search Procedure for Determining $N^*$

To identify the minimal sample size $N^*$ required for a model to successfully learn a target task, we use a hybrid Binary-Geometric search as described in Algorithm 1. The algorithm conducts a search over sample sizes, combining an initial exponential reduction phase with a subsequent binary search phase.

The search begins at a predefined maximum sample size $N_{\text{max}}$. For any candidate size $N$, the algorithm trains models using multiple configurations drawn from a fixed hyperparameter grid $\Phi$. In our implementation, each evaluation consists of 15 model instances (3 learning rates $\times$ 5 random seeds). A sample size $N$ is considered successful if at least one configuration attains validation loss below a threshold $\epsilon$, in which case we decrease the next sample size, and otherwise, the size is labeled unsuccessful and the next trial size is increased.

---

**Algorithm 1** Binary–Geometric Search for $N^*$

**Inputs:** Max sample size $N_{\text{max}}$, Geometric Multiplier $M$, Step limit $S$, Hyperparameter grid $\Phi$, threshold $\epsilon$
**Output:** Minimal sample size $N^*$

  1: $L \leftarrow 0$                                                         *# Lower bound (largest failed size tried so far)*
  2: $N^* \leftarrow N_{\text{max}}$                                        *# Best size so far (smallest successful size tried so far)*
  3: $N \leftarrow N_{\text{max}}$                                          *# Current candidate*
  4: $step \leftarrow 0$

  5: **while** $step < S$ **do**

  6:     *# Step 1: Evaluate model configuration at size $N$*
  7:     $Success \leftarrow$ **false**
  8:     **for all** $\phi \in \Phi$ **do**
  9:         $\mathcal{L}_{\text{val}} \leftarrow$ Train and evaluate model with $\phi$ on $N$ samples
10:         **if** $\mathcal{L}_{\text{val}} < \epsilon$ **then**
11:             $Success \leftarrow$ **true**
12:             **break**

13:     *# Step 2: Update bounds & choose next candidate*
14:     **if** $Success$ **then**
15:         $N^* \leftarrow N$                          *# Update the best value*
16:         **if** $L = 0$: $N \leftarrow N \mathbin{/\!/} M$        *# Phase 1: Exponential decay*
17:         **else:** $N \leftarrow (L + N) \mathbin{/\!/} 2$      *# Phase 2: Binary search update*
18:     **else**
19:         **if** $N = N_{\text{max}}$ **then**
20:             **return** $-1$                    *# Failure: task not learned with max sample size*
21:         $L \leftarrow N$                         *# Update lower bound*
22:         $N \leftarrow (N + N^*) \mathbin{/\!/} 2$        *# Binary search update*

23:     $step \leftarrow step + 1$
24: **return** $N^*$

---

We use a geometric multiplier of $M = 1000$, a maximum of $S = 20$ search steps, and a success threshold of $\epsilon = 10^{-4}$. The hyperparameter grid is

$$\Phi = \{\text{LR} \in \{10^{-3}, 10^{-4}, 10^{-5}\}\} \times \{\text{seed} \in \{10, 20, 30, 40, 50\}\},$$

yielding 15 configurations per evaluation. Each model is trained for at most 250k optimization steps with batch size 64 using the Adam optimizer (Kingma & Ba, 2014), independent of the training set size $N$, with early stopping once the validation-loss success criterion is met. This implies a maximum feasible sample size of

$$N_{\text{max}} = 250{,}000 \times 64 = 16\text{M}.$$

The maximum training set size is the minimum between the feasible 16M samples and the total number of sequences

available under the specified configuration of maximum sequence length $L$ and modulus $m$: $m^L$ for the *Fixed* distribution, and $\sum_{n=2}^{L} m^n$ for the uniform and short-to-long distributions (see Section 2). The final training set size is obtained after deducting the validation set.

## A.2. Evaluation

We ensure that the training and validation sets are strictly disjoint. The validation set contains 2,000 samples (or at most 20% of the available data) and remains identical across different training set sizes, except for variations introduced by the random seed. In addition, we always use at most 20% of the available samples at each sequence length for validation, with the remainder reserved exclusively for training. Also, for all tasks, multi-digit integers are represented as single tokens during tokenization. Finally, for the Chain-of-Thought task format, validation loss is computed using teacher forcing rather than autoregressive sampling.

## A.3. Models

The transformer model is based on the GPT-2 architecture (Radford et al., 2019), with 6 layers and a model (embedding/hidden) dimension of 256. Other architectural parameters, including an MLP expansion factor of 4, follow the default GPT-2 (small) settings. The Mamba model used in Section 4 has 6 layers, model dimension 256, expansion factor 2 (inner dimension 512), state size 64, and convolution kernel size 4.

Both the LSTM and Dense-SSM use a single-layer recurrent cell followed by a linear classification head to map the hidden state to token logits. We use an input and hidden dimension of 768 for the LSTM, and 256 for the Dense-SSM. See Ebrahimi & Memisevic (2025) for additional details on the Bilinear architecture used for Dense-SSM. We also experiment with a 2-layer transformer and a single-layer LSTM with a hidden dimensionality of 256; sample-efficiency results for these variants are provided in the Appendix Section B.3.

# B. Additional Experimental Results

## B.1. Evaluating Length Generalization

Table 2 reports accuracy on sequences of length $2\times$ the maximum length used during training, normalized such that 0 corresponds to random chance. All models are trained using the maximum available training set size for each configuration.

*Table 2.* Accuracy on sequences of length $2\times$ the maximum used during training, normalized such that 0 corresponds to random chance.

| Model | Format | Length Modulus | Fixed | | | | Uniform | | | | Short-to-Long | | | |
|---|---|---|---|---|---|---|---|---|---|---|---|---|---|---|
| | | | 5 | 10 | 20 | 30 | 5 | 10 | 20 | 30 | 5 | 10 | 20 | 30 |
| Transformer | Outcome Supervision | 2 | 0.00 | 0.00 | 0.01 | 0.00 | 0.01 | 0.00 | 0.00 | 0.00 | 0.00 | 0.00 | 0.00 | 0.00 |
| | | 3 | 0.01 | 0.00 | 0.01 | 0.00 | 0.00 | 0.04 | 0.00 | 0.00 | 0.00 | 0.03 | 0.00 | 0.00 |
| | | 5 | 0.00 | 0.00 | 0.00 | 0.00 | 0.00 | 0.03 | 0.01 | 0.00 | 0.00 | 0.00 | 0.00 | 0.00 |
| | | 10 | 0.00 | 0.02 | 0.00 | 0.00 | 0.00 | 0.00 | 0.01 | 0.00 | 0.01 | 0.00 | 0.00 | 0.00 |
| | | 15 | 0.00 | 0.01 | 0.00 | 0.00 | 0.00 | 0.00 | 0.01 | 0.00 | 0.01 | 0.01 | 0.01 | 0.00 |
| | | 20 | 0.00 | 0.00 | 0.01 | 0.00 | 0.00 | 0.01 | 0.01 | 0.00 | 0.00 | 0.00 | 0.00 | 0.00 |
| | | 50 | 0.01 | 0.00 | 0.00 | 0.00 | 0.00 | 0.00 | 0.01 | 0.00 | 0.00 | 0.00 | 0.00 | 0.00 |
| | | 75 | 0.00 | 0.00 | 0.00 | 0.00 | 0.00 | 0.00 | 0.00 | 0.00 | 0.00 | 0.00 | 0.00 | 0.00 |
| | | 100 | 0.00 | 0.00 | 0.00 | 0.00 | 0.00 | 0.00 | 0.00 | 0.00 | 0.00 | 0.00 | 0.00 | 0.00 |
| | Chain of Thought | 2 | 0.00 | 0.01 | 0.00 | 0.02 | 0.00 | 0.00 | 0.00 | 0.00 | 0.00 | 0.00 | 0.00 | 0.00 |
| | | 3 | 0.02 | 0.02 | 0.01 | 0.00 | 0.00 | 0.00 | 0.00 | 0.02 | 0.04 | 0.02 | 0.02 | 0.00 |
| | | 5 | 0.01 | 0.00 | 0.00 | 0.01 | 0.00 | 0.00 | 0.01 | 0.01 | 0.00 | 0.00 | 0.00 | 0.00 |
| | | 10 | 0.01 | 0.01 | 0.01 | 0.01 | 0.01 | 0.00 | 0.00 | 0.00 | 0.00 | 0.00 | 0.00 | 0.00 |
| | | 15 | 0.01 | 0.00 | 0.01 | 0.00 | 0.00 | 0.01 | 0.01 | 0.01 | 0.00 | 0.00 | 0.00 | 0.01 |
| | | 20 | 0.00 | 0.00 | 0.00 | 0.00 | 0.00 | 0.00 | 0.00 | 0.01 | 0.00 | 0.00 | 0.01 | 0.00 |
| | | 50 | 0.00 | 0.00 | 0.00 | 0.00 | 0.00 | 0.00 | 0.00 | 0.00 | 0.00 | 0.00 | 0.00 | 0.00 |
| | | 75 | 0.00 | 0.00 | 0.00 | 0.00 | 0.01 | 0.00 | 0.00 | 0.00 | 0.01 | 0.00 | 0.00 | 0.00 |
| | | 100 | 0.00 | 0.00 | 0.00 | 0.00 | 0.00 | 0.00 | 0.00 | 0.00 | 0.00 | 0.00 | 0.00 | 0.00 |
| | Aligned Chain-of-Thought | 2 | 0.00 | 0.00 | 0.00 | 0.00 | 0.01 | 0.00 | 0.00 | 0.00 | 0.01 | 0.02 | 0.01 | 0.00 |
| | | 3 | 0.00 | 0.01 | 0.00 | 0.00 | 0.00 | 0.00 | 0.03 | 0.00 | 0.00 | 0.01 | 0.02 | 0.00 |
| | | 5 | 0.00 | 0.01 | 0.01 | 0.00 | 0.00 | 0.01 | 0.00 | 0.01 | 0.00 | 0.01 | 0.00 | 0.00 |
| | | 10 | 0.00 | 0.01 | 0.00 | 0.00 | 0.00 | 0.00 | 0.00 | 0.00 | 0.01 | 0.01 | 0.01 | 0.00 |
| | | 15 | 0.01 | 0.00 | 0.01 | 0.00 | 0.01 | 0.00 | 0.01 | 0.01 | 0.00 | 0.00 | 0.01 | 0.00 |
| | | 20 | 0.01 | 0.00 | 0.00 | 0.00 | 0.01 | 0.01 | 0.00 | 0.00 | 0.00 | 0.00 | 0.00 | 0.00 |
| | | 50 | 0.00 | 0.00 | 0.00 | 0.00 | 0.00 | 0.00 | 0.00 | 0.00 | 0.00 | 0.00 | 0.00 | 0.00 |
| | | 75 | 0.00 | 0.00 | 0.00 | 0.00 | 0.00 | 0.00 | 0.00 | 0.00 | 0.00 | 0.00 | 0.00 | 0.00 |
| | | 100 | 0.00 | 0.01 | 0.00 | 0.00 | 0.00 | 0.00 | 0.00 | 0.00 | 0.00 | 0.00 | 0.00 | 0.00 |
| LSTM | Outcome Supervision | 2 | 0.09 | 1.00 | 0.60 | 0.00 | 1.00 | 1.00 | 1.00 | 0.02 | 1.00 | 1.00 | 0.99 | 0.00 |
| | | 3 | 0.05 | 0.00 | 0.12 | 0.02 | 1.00 | 1.00 | 0.96 | 0.99 | 1.00 | 0.98 | 0.18 | 0.29 |
| | | 5 | 0.61 | 0.01 | 0.01 | 0.00 | 0.88 | 0.93 | 0.84 | 1.00 | 0.98 | 0.00 | 0.00 | 0.01 |
| | | 10 | 0.13 | 0.01 | 0.01 | 0.01 | 0.86 | 0.59 | 0.99 | 0.13 | 0.26 | 0.12 | 0.12 | 0.12 |
| | | 15 | 0.92 | 0.00 | 0.01 | 0.00 | 0.50 | 0.11 | 0.01 | 0.01 | 0.17 | 0.06 | 0.16 | 0.02 |
| | | 20 | 0.00 | 0.00 | 0.01 | 0.00 | 0.00 | 0.00 | 0.00 | 0.00 | 0.02 | 0.00 | 0.00 | 0.00 |
| | | 50 | 0.00 | 0.00 | 0.00 | 0.00 | 0.00 | 0.00 | 0.00 | 0.00 | 0.00 | 0.00 | 0.00 | 0.00 |
| | | 75 | 0.00 | 0.00 | 0.00 | 0.00 | 0.00 | 0.00 | 0.00 | 0.00 | 0.00 | 0.00 | 0.00 | 0.01 |
| | | 100 | 0.00 | 0.00 | 0.00 | 0.00 | 0.00 | 0.00 | 0.00 | 0.00 | 0.00 | 0.00 | 0.00 | 0.00 |
| | Chain of Thought | 2 | 0.14 | 0.18 | 0.00 | 0.00 | 0.11 | 0.00 | 0.00 | 0.00 | 0.00 | 0.00 | 0.01 | 0.03 |
| | | 3 | 0.03 | 0.00 | 0.00 | 0.01 | 0.03 | 0.00 | 0.00 | 0.00 | 0.00 | 0.01 | 0.03 | 0.00 |
| | | 5 | 0.01 | 0.00 | 0.00 | 0.00 | 0.04 | 0.01 | 0.00 | 0.01 | 0.00 | 0.00 | 0.00 | 0.01 |
| | | 10 | 0.01 | 0.00 | 0.01 | 0.01 | 0.00 | 0.00 | 0.01 | 0.01 | 0.00 | 0.00 | 0.00 | 0.01 |
| | | 15 | 0.00 | 0.00 | 0.01 | 0.00 | 0.00 | 0.00 | 0.01 | 0.01 | 0.00 | 0.00 | 0.01 | 0.00 |
| | | 20 | 0.00 | 0.00 | 0.00 | 0.00 | 0.00 | 0.00 | 0.00 | 0.00 | 0.01 | 0.00 | 0.00 | 0.00 |
| | | 50 | 0.00 | 0.00 | 0.00 | 0.00 | 0.00 | 0.00 | 0.00 | 0.00 | 0.00 | 0.00 | 0.00 | 0.00 |
| | | 75 | 0.00 | 0.00 | 0.00 | 0.00 | 0.00 | 0.00 | 0.00 | 0.00 | 0.00 | 0.00 | 0.00 | 0.00 |
| | | 100 | 0.00 | 0.00 | 0.00 | 0.00 | 0.00 | 0.00 | 0.00 | 0.00 | 0.00 | 0.00 | 0.00 | 0.00 |
| | Aligned Chain-of-Thought | 2 | 1.00 | 1.00 | 1.00 | 1.00 | 1.00 | 1.00 | 1.00 | 1.00 | 1.00 | 1.00 | 1.00 | 1.00 |
| | | 3 | 1.00 | 1.00 | 1.00 | 1.00 | 1.00 | 1.00 | 1.00 | 1.00 | 1.00 | 1.00 | 1.00 | 1.00 |
| | | 5 | 1.00 | 1.00 | 1.00 | 1.00 | 1.00 | 1.00 | 1.00 | 1.00 | 1.00 | 1.00 | 1.00 | 1.00 |
| | | 10 | 1.00 | 1.00 | 1.00 | 1.00 | 1.00 | 1.00 | 1.00 | 1.00 | 1.00 | 1.00 | 1.00 | 1.00 |
| | | 15 | 1.00 | 1.00 | 1.00 | 1.00 | 1.00 | 1.00 | 1.00 | 1.00 | 1.00 | 1.00 | 1.00 | 1.00 |
| | | 20 | 1.00 | 1.00 | 1.00 | 1.00 | 1.00 | 1.00 | 1.00 | 1.00 | 1.00 | 1.00 | 1.00 | 0.59 |
| | | 50 | 1.00 | 1.00 | 1.00 | 1.00 | 1.00 | 1.00 | 1.00 | 0.99 | 1.00 | 0.98 | 0.45 | 0.14 |
| | | 75 | 1.00 | 1.00 | 0.99 | 1.00 | 1.00 | 1.00 | 1.00 | 0.99 | 0.97 | 0.16 | 0.02 | 0.02 |
| | | 100 | 1.00 | 1.00 | 0.99 | 1.00 | 1.00 | 1.00 | 0.99 | 0.98 | 0.95 | 0.16 | 0.05 | 0.03 |
| Dense-SSM | Outcome Supervision | 2 | 0.00 | 1.00 | 1.00 | 1.00 | 1.00 | 1.00 | 1.00 | 1.00 | 1.00 | 1.00 | 1.00 | 1.00 |
| | | 3 | 0.00 | 1.00 | 0.03 | 0.00 | 1.00 | 1.00 | 1.00 | 1.00 | 1.00 | 1.00 | 1.00 | 1.00 |
| | | 5 | 0.00 | 0.00 | 0.01 | 0.00 | 1.00 | 1.00 | 1.00 | 1.00 | 1.00 | 1.00 | 1.00 | 0.29 |
| | | 10 | 1.00 | 0.10 | 0.01 | 0.02 | 1.00 | 1.00 | 1.00 | 0.10 | 1.00 | 0.13 | 0.11 | 0.12 |
| | | 15 | 0.00 | 0.00 | 0.00 | 0.01 | 1.00 | 1.00 | 1.00 | 1.00 | 1.00 | 0.32 | 0.64 | 0.00 |
| | | 20 | 0.47 | 0.00 | 0.01 | 0.01 | 1.00 | 1.00 | 1.00 | 0.48 | 1.00 | 0.00 | 0.02 | 0.00 |
| | | 50 | 0.00 | 0.00 | 0.00 | 0.00 | 1.00 | 0.50 | 0.18 | 0.00 | 1.00 | 0.00 | 0.00 | 0.00 |
| | | 75 | 0.00 | 0.00 | 0.00 | 0.00 | 1.00 | 0.83 | 0.00 | 0.00 | 0.00 | 0.00 | 0.00 | 0.00 |
| | | 100 | 0.00 | 0.00 | 0.00 | 0.00 | 0.65 | 0.25 | 0.00 | 0.00 | 0.00 | 0.00 | 0.00 | 0.00 |
| | Chain of Thought | 2 | 0.00 | 0.03 | 0.01 | 0.00 | 0.00 | 0.00 | 0.00 | 0.00 | 0.00 | 0.02 | 0.02 | 0.01 |
| | | 3 | 0.02 | 0.00 | 0.01 | 0.02 | 0.00 | 0.00 | 0.00 | 0.01 | 0.02 | 0.02 | 0.00 | 0.01 |
| | | 5 | 0.00 | 0.00 | 0.00 | 0.00 | 0.00 | 0.00 | 0.00 | 0.00 | 0.00 | 0.00 | 0.00 | 0.00 |
| | | 10 | 0.01 | 0.00 | 0.00 | 0.00 | 0.00 | 0.00 | 0.00 | 0.00 | 0.00 | 0.00 | 0.00 | 0.00 |
| | | 15 | 0.00 | 0.01 | 0.01 | 0.00 | 0.00 | 0.00 | 0.00 | 0.01 | 0.00 | 0.00 | 0.00 | 0.01 |
| | | 20 | 0.00 | 0.00 | 0.00 | 0.00 | 0.00 | 0.00 | 0.00 | 0.00 | 0.00 | 0.01 | 0.01 | 0.00 |
| | | 50 | 0.00 | 0.00 | 0.00 | 0.00 | 0.00 | 0.00 | 0.00 | 0.01 | 0.00 | 0.00 | 0.00 | 0.00 |
| | | 75 | 0.00 | 0.00 | 0.00 | 0.00 | 0.00 | 0.00 | 0.00 | 0.00 | 0.00 | 0.00 | 0.00 | 0.00 |
| | | 100 | 0.00 | 0.00 | 0.00 | 0.00 | 0.00 | 0.00 | 0.00 | 0.00 | 0.00 | 0.00 | 0.00 | 0.00 |
| | Aligned Chain-of-Thought | 2 | 1.00 | 1.00 | 1.00 | 1.00 | 1.00 | 1.00 | 1.00 | 1.00 | 1.00 | 1.00 | 1.00 | 1.00 |
| | | 3 | 1.00 | 1.00 | 1.00 | 1.00 | 1.00 | 1.00 | 1.00 | 1.00 | 1.00 | 1.00 | 1.00 | 1.00 |
| | | 5 | 1.00 | 1.00 | 1.00 | 1.00 | 1.00 | 1.00 | 1.00 | 1.00 | 1.00 | 1.00 | 1.00 | 1.00 |
| | | 10 | 1.00 | 1.00 | 1.00 | 1.00 | 1.00 | 1.00 | 1.00 | 1.00 | 1.00 | 1.00 | 1.00 | 1.00 |
| | | 15 | 1.00 | 1.00 | 1.00 | 1.00 | 1.00 | 1.00 | 1.00 | 1.00 | 1.00 | 1.00 | 1.00 | 0.93 |
| | | 20 | 1.00 | 1.00 | 1.00 | 1.00 | 1.00 | 1.00 | 1.00 | 1.00 | 1.00 | 1.00 | 1.00 | 1.00 |
| | | 50 | 1.00 | 1.00 | 1.00 | 1.00 | 1.00 | 1.00 | 1.00 | 1.00 | 1.00 | 1.00 | 1.00 | 1.00 |
| | | 75 | 1.00 | 1.00 | 1.00 | 1.00 | 1.00 | 1.00 | 1.00 | 1.00 | 1.00 | 1.00 | 1.00 | 1.00 |
| | | 100 | 1.00 | 1.00 | 1.00 | 1.00 | 1.00 | 1.00 | 1.00 | 1.00 | 1.00 | 1.00 | 1.00 | 1.00 |

## B.2. Permutation Composition Task

**Task:** To show our findings generalize beyond commutative operations, we consider the task of permutation composition (simulating the symmetric group $S_m$). Each element of the group represents a permutation of the set $\{1, \ldots, m\}$, resulting in a group of cardinality $|S_m| = m!$. In our experimental setup, each permutation $\pi \in S_m$ is bijectively mapped to a unique integer token in $\{0, 1, \ldots, m! - 1\}$. Given an input sequence of $n$ permutations $\mathbf{x} = (\pi_1, \pi_2, \ldots, \pi_n)$, the model is required to compute their sequential composition:

$$y = \pi_n \circ \pi_{n-1} \circ \cdots \circ \pi_1, \tag{5}$$

where $\circ$ denotes the permutation composition operator. This task significantly elevates the complexity of state tracking, as the model can no longer rely on the order-invariance property characteristic of abelian groups.

**Algebraic Significance:** The symmetric group $S_m$ serves as the canonical non-commutative structure for evaluating state tracking. Its fundamental importance is grounded in *Cayley's Theorem*, which states that every finite group $G$ is isomorphic to a subgroup of the symmetric group $S_{|G|}$ (Dummit & Foote, 2003). Hence, by analyzing performance on $S_m$, we effectively probe the model's capacity to internalize the transition dynamics of any finite discrete group.

As noted in Figure 7, we observe the same patterns described in Section 4 and Figure 6, supporting the generalization of these findings and the subsequent arguments to non-commutative state-tracking tasks.

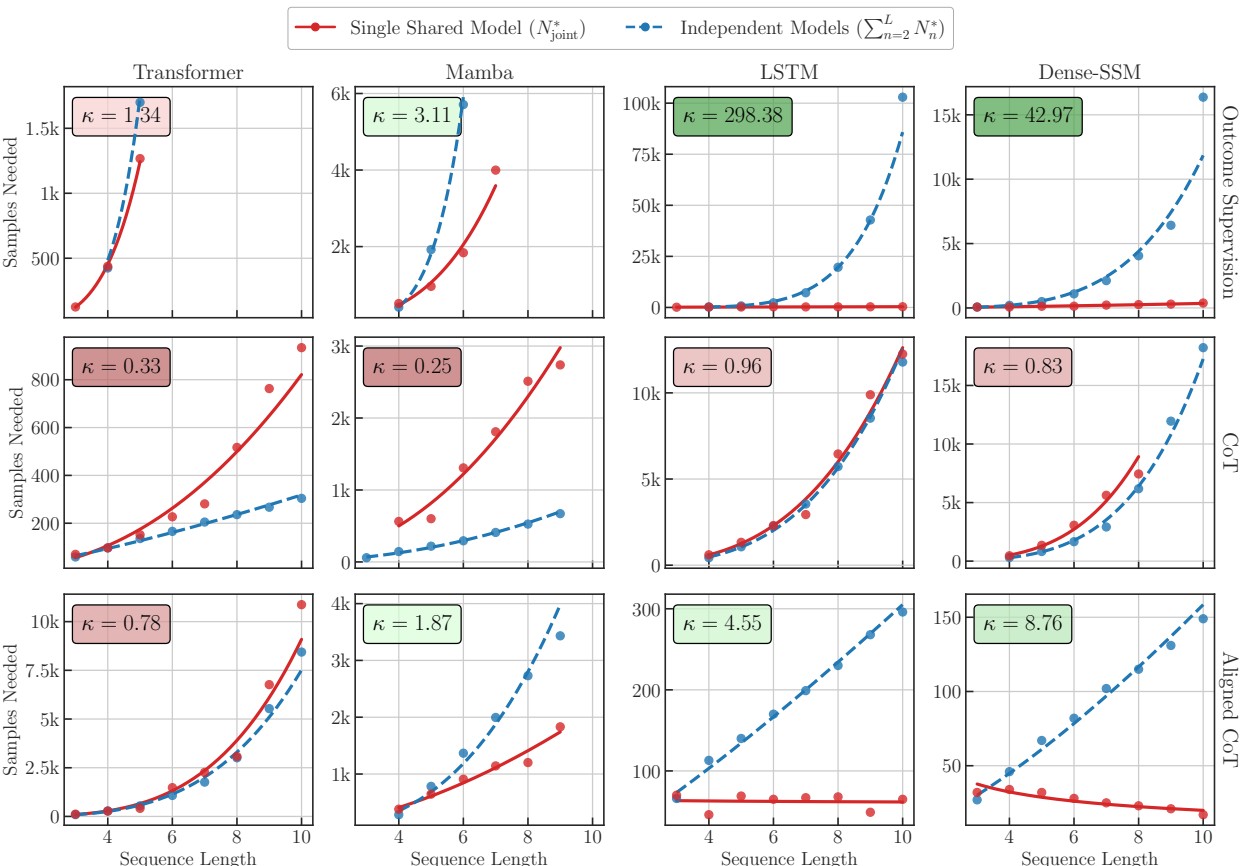

*Figure 7.* Similar to Figure 6, but for permutation composition task ($S_5$). The results suggest that transformers, and to a lesser extent Mamba, learn largely isolated solutions for each sequence length.

## B.3. Data Efficiency Evaluation for Smaller Models

Table 3. $N^*$ for LSTMs with 256 and 768 hidden dimensions. We observe similar trends across both model sizes.

| Model | Format | Length Modulus | Fixed 5 | 10 | 20 | 30 | Uniform 5 | 10 | 20 | 30 | Short-to-Long 5 | 10 | 20 | 30 |
|---|---|---|---|---|---|---|---|---|---|---|---|---|---|---|
| LSTM Dim = 256 | Outcome Supervision | 2 | 14 | 202 | 257 | 413 | 26 | 34 | 67 | 109 | 12 | 13 | 26 | 16 |
| | | 3 | 151 | 620 | – | – | 84 | 122 | 317 | 686 | 81 | 180 | 132 | 178 |
| | | 5 | 1.3K | 6.7K | – | – | 487 | 876 | – | – | 788 | 1.4K | – | – |
| | | 10 | 6.3K | – | – | – | 3.4K | – | – | – | 5.1K | – | – | – |
| | | 15 | 21.9K | – | – | – | 11.1K | – | – | – | – | – | – | – |
| | | 20 | – | – | – | – | – | – | – | – | – | – | – | – |
| | | 50 | – | – | – | – | – | – | – | – | – | – | – | – |
| | | 75 | – | – | – | – | – | – | – | – | – | – | – | – |
| | | 100 | – | – | – | – | – | – | – | – | – | – | – | – |
| | Chain of Thought | 2 | 26 | 346 | 3.2K | 14.9K | – | 853 | 24.6K | 306.4K | – | 920 | 581.1K | – |
| | | 3 | – | 5K | 20.5K | – | – | 12.2K | 129.2K | – | – | 29.9K | – | – |
| | | 5 | – | 9.6K | 32K | – | – | 26.3K | 68.1K | – | – | 2.5M | – | – |
| | | 10 | 10.4K | 14.8K | 15.4M | – | 21.7K | 53.1K | – | – | 21.6K | – | – | – |
| | | 15 | 19.3K | 21.1K | – | – | 46.3K | 162.1K | – | – | 73.7K | – | – | – |
| | | 20 | 30.9K | 21.6K | – | – | 64.6K | 351.8K | – | – | 276.7K | – | – | – |
| | | 50 | 33.7K | 36.5K | – | – | 413.8K | 14.7M | – | – | 8.4M | – | – | – |
| | | 75 | 41.1K | 50.5K | – | – | 595.1K | – | – | – | – | – | – | – |
| | | 100 | 46.4K | 13.4M | – | – | 3.8M | – | – | – | – | – | – | – |
| | Aligned Chain-of-Thought | 2 | 8 | 8 | 10 | 8 | 4 | 8 | 2 | 8 | 12 | 7 | 10 | 8 |
| | | 3 | 10 | 11 | 8 | 8 | 21 | 17 | 8 | 8 | 27 | 32 | 39 | 39 |
| | | 5 | 95 | 75 | 52 | 39 | 144 | 101 | 70 | 54 | 142 | 148 | 225 | 209 |
| | | 10 | 324 | 227 | 205 | 209 | 477 | 379 | 317 | 257 | 606 | 1.2K | 1.7K | 1.9K |
| | | 15 | 543 | 502 | 452 | 467 | 846 | 677 | 637 | 772 | 1.4K | 4K | 8.4K | 57.6K |
| | | 20 | 993 | 902 | 919 | 1K | 1.2K | 1.3K | 1.2K | 1.2K | 2.6K | 10.1K | 176.3K | 194.9K |
| | | 50 | 4.6K | 5.3K | 7.3K | 4K | 5.8K | 7.3K | 8.7K | 11.5K | 20K | 168.2K | – | – |
| | | 75 | 9.2K | 9.2K | 13.5K | 15.4K | 10.3K | 12.2K | 15.7K | 20K | 42.6K | 1.2M | – | – |
| | | 100 | 14.5K | 15.6K | 19.8K | 32.6K | 16K | 23.9K | 25.3K | 9.1M | 78.6K | – | – | – |
| LSTM Dim = 768 | Outcome Supervision | 2 | 14 | 101 | 252 | 309 | 16 | 21 | 64 | 54 | 12 | 9 | 13 | 16 |
| | | 3 | 195 | 620 | 1.7K | – | 78 | 122 | 317 | 526 | 83 | 90 | 90 | 100 |
| | | 5 | 2K | 5.9K | – | – | 387 | 707 | – | – | 381 | 22.7K | – | – |
| | | 10 | 6.4K | – | – | – | 2.2K | 8.6K | – | – | 2.2K | – | – | – |
| | | 15 | 18.8K | – | – | – | 11.8K | – | – | – | – | – | – | – |
| | | 20 | – | – | – | – | – | – | – | – | – | – | – | – |
| | | 50 | – | – | – | – | – | – | – | – | – | – | – | – |
| | | 75 | – | – | – | – | – | – | – | – | – | – | – | – |
| | | 100 | – | – | – | – | – | – | – | – | – | – | – | – |
| | Chain of Thought | 2 | 20 | 307 | 4.7K | 8.6K | 45 | 692 | 13.4K | 37.1K | 48 | 843 | 1.1M | – |
| | | 3 | – | 2.9K | 9.3K | 14K | – | 7.4K | 40.3K | 5.2M | – | 29K | – | – |
| | | 5 | 1.6K | 10.3K | 20.4K | 44K | 2K | 28.4K | 68.1K | 2.1M | 2.1K | – | – | – |
| | | 10 | 6.1K | 15.7K | 29.1K | – | 14.3K | 544.4K | 500K | – | 16.8K | – | – | – |
| | | 15 | 15.8K | 21.1K | 42K | – | 32K | 67K | – | – | 65.6K | – | – | – |
| | | 20 | 21.2K | 29.6K | 54.3K | – | 40.7K | 409.1K | 14M | – | 215.4K | – | – | – |
| | | 50 | 43K | 50.4K | – | – | 206K | 2.2M | – | – | 10.4M | – | – | – |
| | | 75 | 56.5K | 58K | – | – | 4M | – | – | – | – | – | – | – |
| | | 100 | 84.2K | 9.5M | – | – | – | – | – | – | – | – | – | – |
| | Aligned Chain-of-Thought | 2 | 4 | 6 | 2 | 2 | 4 | 4 | 2 | 2 | 9 | 7 | 7 | 8 |
| | | 3 | 10 | 8 | 8 | 6 | 18 | 12 | 6 | 8 | 27 | 28 | 34 | 34 |
| | | 5 | 95 | 70 | 39 | 31 | 153 | 107 | 80 | 47 | 142 | 148 | 253 | 235 |
| | | 10 | 278 | 194 | 146 | 130 | 447 | 313 | 247 | 232 | 685 | 1.3K | 1.8K | 2K |
| | | 15 | 534 | 478 | 409 | 371 | 865 | 631 | 567 | 500 | 1.5K | 3.9K | 5.2K | 7.1K |
| | | 20 | 798 | 672 | 652 | 798 | 1.3K | 934 | 870 | 820 | 2.7K | 9.4K | 13.4K | 25.9K |
| | | 50 | 3.7K | 4.6K | 6K | 7.4K | 4.1K | 5.3K | 6.1K | 6.9K | 14.7K | 156.2K | – | – |
| | | 75 | 7.4K | 9.2K | 12.2K | 13.3K | 8.4K | 11.3K | 14.2K | 16.8K | 41.1K | 697.4K | – | – |
| | | 100 | 12.8K | 16.6K | 21.9K | 23.7K | 13.3K | 19.4K | 24.3K | 28.5K | 99.7K | – | – | – |

Table 4. $N^*$ for transformers with 2 and 6 layers. We observe similar trends across both model depths.

| Model | Format | Length Modulus | Fixed 5 | 10 | 20 | 30 | Uniform 5 | 10 | 20 | 30 | Short-to-Long 5 | 10 | 20 | 30 |
|---|---|---|---|---|---|---|---|---|---|---|---|---|---|---|
| Transformer 2 Layers | Outcome Supervision | 2 | – | 594 | – | – | 48 | 807 | – | – | 48 | 971 | – | – |
| | | 3 | 134 | – | – | – | – | – | – | – | – | – | – | – |
| | | 5 | 1.5K | – | – | – | 1.7K | – | – | – | – | – | – | – |
| | | 10 | – | – | – | – | – | – | – | – | – | – | – | – |
| | | 15 | – | – | – | – | – | – | – | – | – | – | – | – |
| | | 20 | – | – | – | – | – | – | – | – | – | – | – | – |
| | | 50 | – | – | – | – | – | – | – | – | – | – | – | – |
| | | 75 | – | – | – | – | – | – | – | – | – | – | – | – |
| | | 100 | – | – | – | – | – | – | – | – | – | – | – | – |
| | Chain of Thought | 2 | 14 | 26 | 21 | 23 | 37 | 214 | 724 | 1.1K | 42 | 824 | 1M | – |
| | | 3 | 19 | 24 | 25 | 31 | 93 | 456 | 1.3K | 1.7K | 117 | 28.8K | – | – |
| | | 5 | 32 | 31 | 34 | 39 | 241 | 893 | 1.8K | 2K | 653 | 2.4M | – | – |
| | | 10 | 86 | 61 | 52 | 54 | 600 | 1.8K | 3.6K | 4.1K | 11.1K | – | – | – |
| | | 15 | 160 | 97 | 79 | 62 | 805 | 2.9K | 7.4K | 6.8K | 54.2K | – | – | – |
| | | 20 | 229 | 142 | 97 | 78 | 1.3K | 3K | 8.5K | 10K | 168.8K | – | – | – |
| | | 50 | 795 | 542 | 334 | 272 | 3.2K | 6.2K | – | 16M | 6.5M | – | – | – |
| | | 75 | 1.5K | 958 | 688 | 600 | 5.1K | 15.1K | 70.8K | – | – | – | – | – |
| | | 100 | 2.1K | 1.5K | 1.1K | 1K | 6.8K | 20.3K | 33.6K | 209.2K | – | – | – | – |
| | Aligned Chain-of-Thought | 2 | 26 | 197 | 3K | 2K | 37 | 439 | 5.2K | 363.8K | 36 | 869 | 1.1M | – |
| | | 3 | 129 | 2.1K | 7.5K | – | 239 | 11.1K | – | – | 234 | 29.9K | – | – |
| | | 5 | 1.1K | 6.5K | – | – | 1.6K | 40.9K | – | – | 1.5K | 2.5M | – | – |
| | | 10 | 2.1K | 12.4K | – | – | 4.6K | 156.1K | – | – | 11.9K | – | – | – |
| | | 15 | 3.1K | 31.2K | – | – | 8.2K | 1.6M | – | – | 55.3K | – | – | – |
| | | 20 | 4.2K | 59.5K | – | – | 11.2K | – | – | – | 175.5K | – | – | – |
| | | 50 | 10.1K | – | – | – | 15.8K | – | – | – | 6.8M | – | – | – |
| | | 75 | 21.6K | – | – | – | 33.8K | – | – | – | – | – | – | – |
| | | 100 | 25K | – | – | – | 99.8K | – | – | – | – | – | – | – |
| Transformer 6 Layers | Outcome Supervision | 2 | 19 | 364 | – | – | 37 | 549 | – | – | 45 | 913 | – | – |
| | | 3 | 119 | – | – | – | – | – | – | – | 243 | – | – | – |
| | | 5 | 1.1K | – | – | – | 2.6K | – | – | – | 1.5K | – | – | – |
| | | 10 | – | – | – | – | – | – | – | – | – | – | – | – |
| | | 15 | – | – | – | – | – | – | – | – | – | – | – | – |
| | | 20 | – | – | – | – | – | – | – | – | – | – | – | – |
| | | 50 | – | – | – | – | – | – | – | – | – | – | – | – |
| | | 75 | – | – | – | – | – | – | – | – | – | – | – | – |
| | | 100 | – | – | – | – | – | – | – | – | – | – | – | – |
| | Chain of Thought | 2 | 10 | 16 | 19 | 20 | 36 | 198 | 744 | 1K | 39 | 824 | 1M | – |
| | | 3 | 16 | 18 | 21 | 25 | 78 | 465 | 1.2K | 1.4K | 108 | 28.8K | – | – |
| | | 5 | 33 | 30 | 31 | 34 | 148 | 1.1K | 1.7K | 1.7K | 647 | 2.4M | – | – |
| | | 10 | 94 | 66 | 62 | 70 | 427 | 2.3K | 3.4K | 4K | 11K | – | – | – |
| | | 15 | 166 | 116 | 78 | 107 | 709 | 2.5K | 5K | 5.1K | 54.3K | – | – | – |
| | | 20 | 377 | 178 | 116 | 135 | 1K | 3.8K | 5.6K | 7.4K | 168.6K | – | – | – |
| | | 50 | 1.1K | 678 | 553 | 470 | 2.6K | 8.6K | 12K | 11.2K | 6.4M | – | – | – |
| | | 75 | 1.7K | 1.2K | 1K | 946 | 4.2K | 11K | 18.1K | 16.1K | – | – | – | – |
| | | 100 | 2.5K | 1.9K | 1.6K | 1.6K | 5.8K | 13.2K | 21.1K | 19.1K | – | – | – | – |
| | Aligned Chain-of-Thought | 2 | 16 | 121 | 644 | 1.2K | 27 | 313 | 2K | 11.7K | 19 | 856 | 1M | – |
| | | 3 | 91 | 2K | 3.7K | 4K | 180 | 8.3K | 66.8K | – | 207 | 29.7K | – | – |
| | | 5 | 528 | 3.9K | 9.2K | 20.8K | 671 | 14.5K | 2M | – | 909 | 2.4M | – | – |
| | | 10 | 1.3K | 6K | 15.5K | 131.4K | 1.7K | 32.3K | – | – | 11.2K | – | – | – |
| | | 15 | 1.5K | 7.9K | 23.3K | – | 2.4K | 61.5K | – | – | 55K | – | – | – |
| | | 20 | 2.6K | 21.9K | 85.3K | – | 4.5K | 190.9K | – | – | 172.6K | – | – | – |
| | | 50 | 13K | 1M | – | – | 22K | – | – | – | 6.7M | – | – | – |
| | | 75 | 21.1K | 15.2M | – | – | 102.1K | – | – | – | – | – | – | – |
| | | 100 | 23.6K | – | – | – | 8M | – | – | – | – | – | – | – |

## B.4. Weight Sharing in Additional Transformer Variants

To test whether the observed cross-length sharing behavior is specific to the GPT-2-style transformer used in the main experiments, we additionally evaluate transformer variants based on Llama 3 (Grattafiori et al., 2024) and Pythia (Biderman et al., 2023). These variants match the main GPT-2-style transformer in depth, hidden size, number of attention heads, vocabulary size, and effective MLP intermediate size (6 layers, hidden size 256, 4 attention heads, vocabulary size 10, intermediate size 1024), while using more modern architectural choices such as rotary positional embeddings. As shown in Figures 8 and 9, we observe qualitatively similar sharing-factor patterns.

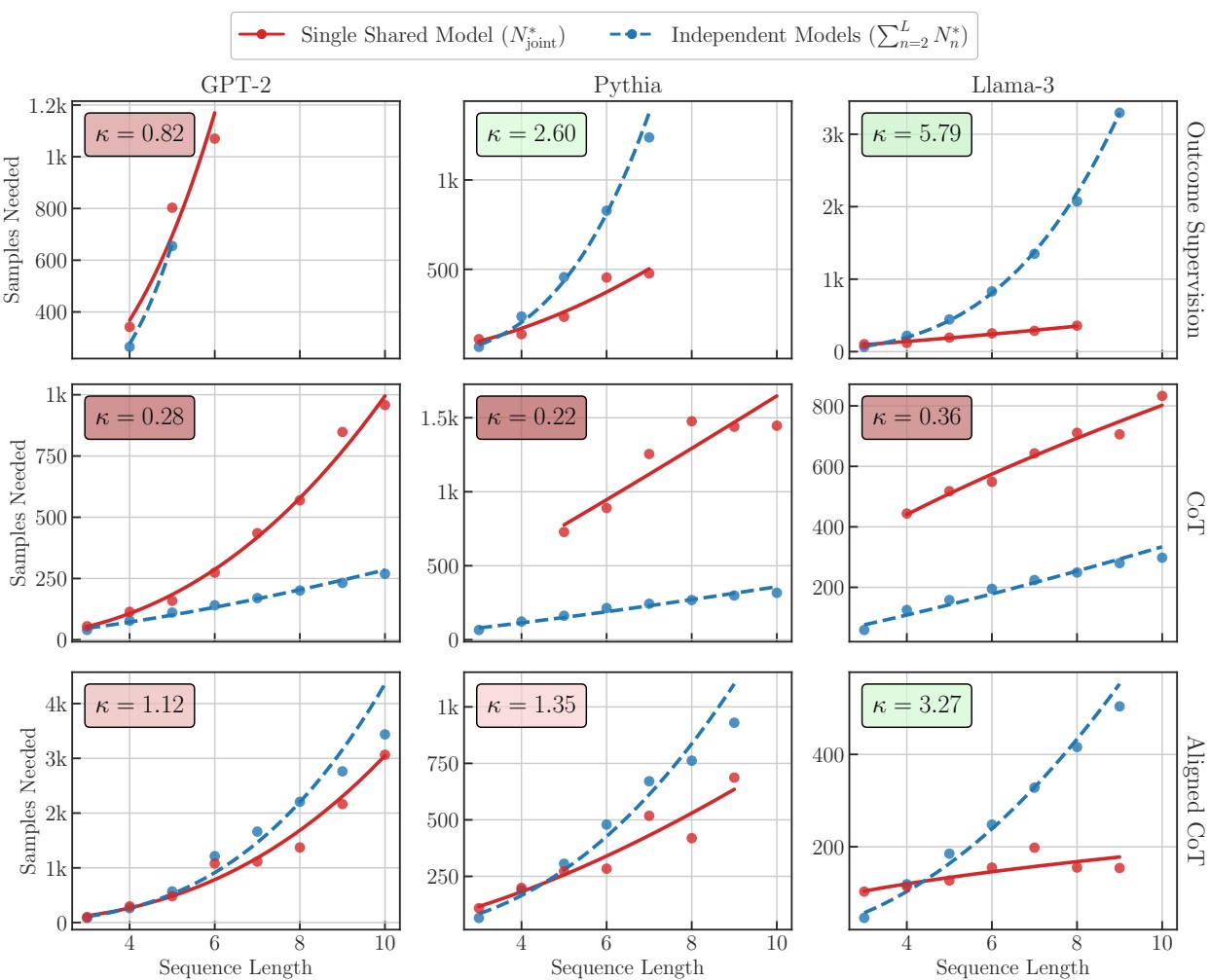

*Figure 8.* Sharing-factor analysis for additional transformer variants on modular addition.

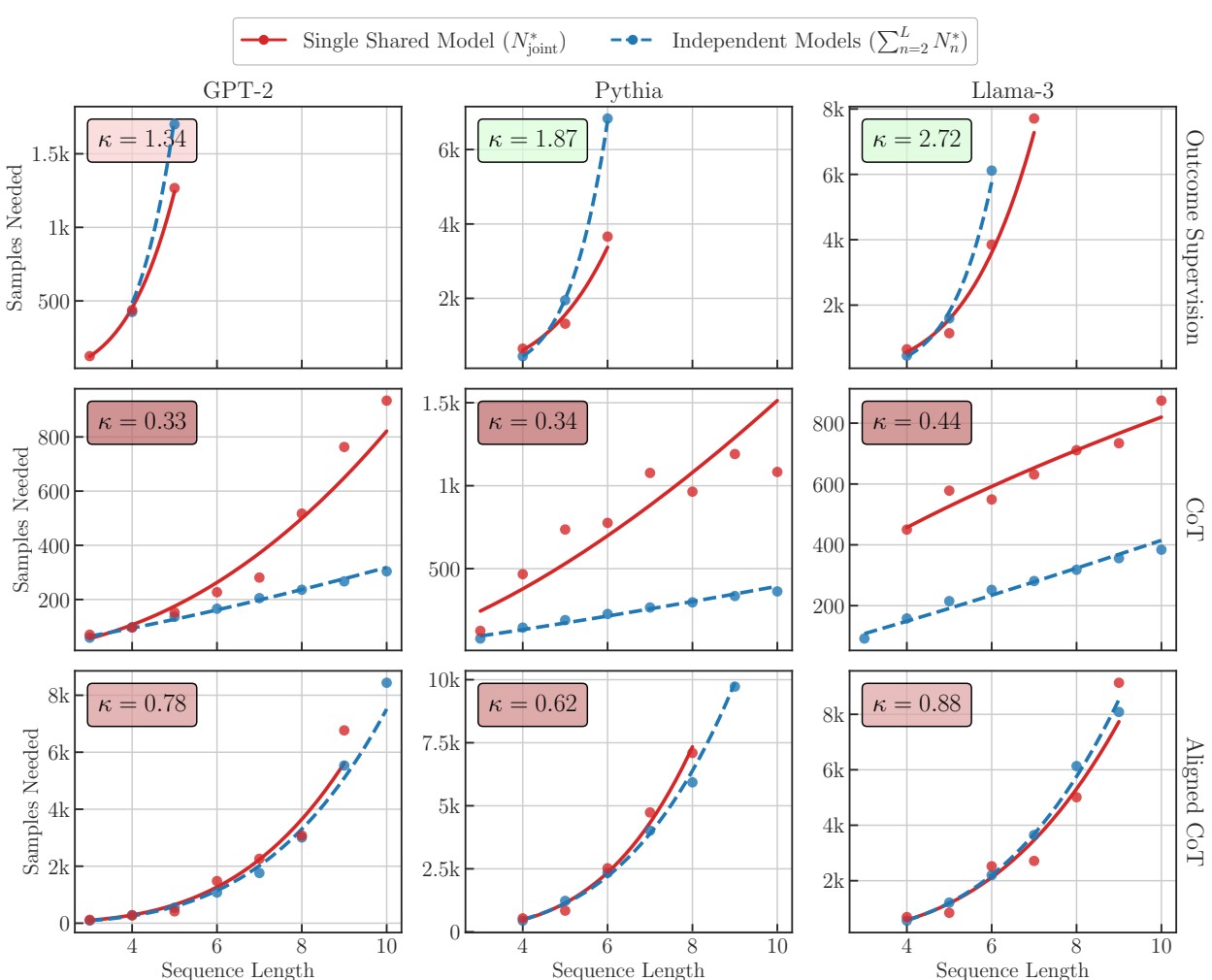

*Figure 9.* Sharing-factor analysis for additional transformer variants on permutation composition over $S_5$.

