# OpenReview forum: "On the "Induction Bias" in Sequence Models"
_ICML.cc/2026/Conference — ICML 2026 regular_

### Official Review · Reviewer_fJCE · 2026-03-07

**Soundness:** 4
**Presentation:** 3
**Significance:** 3
**Originality:** 3
**Overall Recommendation:** 5
**Confidence:** 2

**Summary:**

The paper investigated the data efficiency of transformers in comparison with recurrent neural networks on the empirical level. In specific, they found that with intermediate supervision, the recurrent models require less data than the transformers did. Also, the recurrent models demand fewer samples in the short-to-long setting than in the uniform setting, whereas that phenomenon does not happen on transformers. What's more, they also found that the number of samples to train recurrent networks on all sequence length is smaller than the sum of sample sizes for training on each individual sequence length, indicating that recurrent networks internalize the nature of tasks. In contrast, the transformers prefer to fix length training.

**Compliance With Llm Reviewing Policy:**

Affirmed.

**Final Justification:**

The paper resolves my concerns during the rebuttal phase, and I am willing to raise my scores.

**Key Questions For Authors:**

Thanks for the presentation. I just have a few clarifying questions.

1. Does the validation data have the same length distribution as the training set, ex a model trained under short-to-long setting is validated using data with ascending length?

2. Can you explain intuitively why transformers do not benefit from short-to-long?

3. The last question is about experimental setting in Section 4. For training on the joint task, do you first train and validate on one sequence length, and use the resulting model to train and validate on the next sequence length and so on?

**Limitations:**

yes

**Strengths And Weaknesses:**

The paper investigates the 'induction bias' phenomenon on on two sequence models, transformers and recurrent neural networks. They did extensive experiments to support their claim that recurrent models shares weights across different length, while transformers prefer length-specific training.

---

> ### Author Rebuttal · Authors · 2026-03-31
>
> We thank the reviewer for their time and thoughtful comments and respond below.
>
> > 1. Does the validation data have the same length distribution as the training set, ex a model trained under short-to-long setting is validated using data with ascending length?
>
> Both the training and validation samples are drawn from the same underlying data‑generation process. In the fixed‑length setting, models are trained and evaluated on sequences of the same fixed length. For the uniform and short‑to‑long settings, the validation set includes samples spanning the full range of training lengths, since the ordering of samples during evaluation is irrelevant.
>
> >2. Can you explain intuitively why transformers do not benefit from short-to-long?
>
> Thank you for this question. We consider this result as evidence for the induction bias (or lack thereof, in the case for transformers):
> The induction bias would encourage the step-by-step application of a repeatable atomic operation across the sequence,  allowing a model to generalize from short to long sequences (length generalization).
> This, paired with the fact that it is generally easier to learn from short sequences (better gradient estimates, etc.), could therefore explain the benefits of short-to-long training for RNNs (but not transformers).
>
> >3. The last question is about experimental setting in Section 4. For training on the joint task, do you first train and validate on one sequence length, and use the resulting model to train and validate on the next sequence length and so on?
>
> No, for the joint setting, we train a single model on all sequence lengths simultaneously, similar to the Uniform setting in Section 3. We will clarify this in the final version.

---

> > ### Author Rebuttal · Reviewer_fJCE · 2026-04-03
> >
> > I would like to thank the author for addressing my questions.

---

### Official Review · Reviewer_4g98 · 2026-03-12

**Soundness:** 2
**Presentation:** 2
**Significance:** 2
**Originality:** 2
**Overall Recommendation:** 4
**Confidence:** 4

**Summary:**

This work compares two types of architectures (Transformers and RNN) on the task of modular addition. The analysis focuses on data efficiency, i.e. how many sequences are needed to achieve good prediction, and generalization over sequence length.  Further experiments are done to measure to which extent each architecture tends to share parameters or develop length-specific strategies.

**Compliance With Llm Reviewing Policy:**

Affirmed.

**Ethical Review Flag:**

Flag this paper for an ethics review.

**Final Justification:**

While I have some reserve on the generality of the synthetic tasks, I believe this work can benefit the ICML community.

**Key Questions For Authors:**

None

**Limitations:**

yes

**Strengths And Weaknesses:**

I have some difficulties to relate the very broad scope of "state tracking" as defined in the introduction and the very specific nature of the task of modular addition. On a similar note, I also find the names of the supervision strategies ("(Aligned) Chain-of-Thoughts") very confusing as the potential link between the experiments done in this work and the actual Chain-of-Thought prompting techniques is very brittle.

The results show that RNNs are much more suited than Transformers to learn operations such as modular addition. This is not very surprising as the dependencies in the sequences for this task are strictly left-to-right which aligns well with the RNN architecture. However,  on an arbitrary task of "state tracking" using natural language  where dependencies are much more intricate, I seriously doubt that the insights presented in this work will generalize.

The quality of the presented figures do no meet scientific rigour: the data is way to little to support the *non-linear* tendency lines shown in figures 2, 3, 4, and 5. To give a concrete example: in figure 2, without extra information, two points are not sufficient to extrapolate a quadratic law.

Suggestions:
- The manuscript should be more precise about its scope, i.e. experiments on only modular addition is not enough to derive claim on "state tracking". A statement like:  "a comparison between two LM architectures to learn cyclic operations" would have been more in line with the reported work
- The quality of the figures should be address to meet scientific standards.

---

> ### Author Rebuttal · Authors · 2026-03-31
>
> We thank the reviewer for their time and thoughtful comments and respond below.
>
> > very specific nature of modular addition.
>
> We would like to point out that commutative state tracking tasks, such as Parity and Addition, are commonly used in the literature on studying state tracking capabilities. We do agree that increasing the scope of the state tracking tasks would be interesting, albeit being out of the scope of this (already computationally very demanding) study.
>
> Despite its simplicity, modular addition (cyclic group $\mathbb{Z}_m$) is the canonical commutative state‑tracking task, since every finite abelian group is isomorphic to a direct product of cyclic groups. Additionally, we do use the permutation‑composition task (symmetric group $S_5$) for the weight‑sharing experiments, and reported similar results. It is important to note that permutation composition is representative of a vast range of state tracking tasks, as it structurally encompasses all finite groups.
>
> > link to actual CoT prompting techniques
>
> We use "Chain-of-Thought" (CoT) to refer to intermediate tokens that are explicitly present as training supervision toward a final solution. While the term was originally introduced as an inference-time prompting technique, its use to describe training formats with supervised intermediate steps has also become common in the literature [1, 2, 3]. Because of this, we also use the term to describe task formats where intermediate steps, such as cumulative modular sums, are explicitly present.
>
> [1] Yao et al. Compositional Generalization from Learned Skills via CoT Training: A Theoretical and Structural Analysis for Reasoning, 2025.
>
> [2] Zhang et al. Finite State Automata Inside Transformers with Chain-of-Thought, 2025
>
> [3] Feng et al. Towards a Theory of Learning with Autoregressive Chain of Thought, 2025.
>
> > strictly left-to-right dependencies aligns well with the RNN architecture
>
> We would like to clarify the following points:
>
> 1. Similar to the related recent work on state tracking capabilities of models, our results focus on pure state tracking rather than general natural language. State-tracking in this line of work is generally defined as solving problems over algebraic structures such as groups (see, for example [1-5]). In this work, we use cyclic groups (parity and modular addition) and symmetric groups (permutation composition), both of which are commonly used in the literature.
>
> We can formulate state tracking as a task in which: (1) there is a latent world state, (2) a sequence of updates that are applied through inputs, and (3) the task of finding the final world state based on composing the observed updates.
> In group-theoretic formulations used in this work as well as prior work, this corresponds to defining a group $(G, \circ)$ and computing the cumulative product of the sequence of group elements: $g_1 \circ g_2 \circ \cdots \circ g_T \in G.$
> This abstracts the essence of state tracking: each update to a world state can be modeled as an element of some algebraic structure, and applying updates sequentially corresponds to multiplying those elements.
>
> Therefore, the tasks used in the paper are not intended as merely toy benchmarks. Rather they represent core abstractions of sequential dependencies in temporal data. For example, [1] shows that $S_5$ can be reduced to chess state tracking, and highlights connections to code execution.
>
> 2. It is important to note that the Transformer decoder is also a strictly causal, left-to-right model. Despite sharing the same causal constraints as the RNN, the Transformer fails to internalize a transition operator as a length-invariant mechanism, as evidenced by our low Sharing Factor.
>
> [1] Merrill, W. et al. The Illusion of State in State‑Space Models. ICML, 2024.
>
> [2] Grazzi, et al. Unlocking State-Tracking in Linear RNNs Through Negative Eigenvalues. ICLR 2025.
>
> [3] Siems, et al. DeltaProduct: Improving State-Tracking in Linear RNNs via Householder Products. NeurIPS 2025
>
> [4] Terzić, et al. Structured Sparse Transition Matrices to Enable State Tracking in State-Space Models. NeurIPS 2025
>
> [5] Movahedi, et al. Fixed-Point RNNs: Interpolating from Diagonal to Dense, NeurIPS 2025.
>
> > non-linear tendency lines
>
> We thank the reviewer for catching this. In Figure 2, the y‑axis range was capped at 10K. While the regression lines were fit using all data points reported in Table 1 (and not only the two visible points), the restricted axis range made this unclear. We have revised the figure to display the full y‑axis range and to explicitly plot all available data points.
>
> The regression lines in all figures are intended solely as visual guides, and we do not draw any quantitative or predictive conclusions from them. To improve visual clarity and to avoid misleading extrapolation (e.g., negative values outside the observed data range), we have updated the regression curves across the paper to use a poly‑logarithmic fit.

---

> > ### Author Rebuttal · Reviewer_4g98 · 2026-04-03
> >
> > The authors have adequately answer my comments. I'll increase my score.

---

> > > ### Author Response · Authors · 2026-04-07
> > >
> > > Thank you for your positive response and for indicating you would increase your score. We noticed the score on OpenReview may not have been updated yet. Could you kindly update it when you get a chance? We greatly appreciate your time and engagement.

---

### Official Review · Reviewer_HvxJ · 2026-03-13

**Soundness:** 2
**Presentation:** 3
**Significance:** 2
**Originality:** 2
**Overall Recommendation:** 4
**Confidence:** 3

**Summary:**

This paper presents an empirical study of in-distribution state tracking tasks using Transformers, RNNs, and dense SSMs. The study shows that Transformers are data-inefficient compared to RNNs & SSMs, since they require more training samples as sequence length and state size increase. The study attributes this to an inherent induction bias in RNNs & SSMs towards sharing learned mechanisms across sequence lengths.

**Compliance With Llm Reviewing Policy:**

Affirmed.

**Final Justification:**

I like to thank the authors for the consideration and the rebuttal. I think adding state tracking to the paper is a valuable addition. I will maintain my positive score.

**Key Questions For Authors:**

1. The results regarding CoT and aligned CoT suggest that RNNs & SSMs at large scale should be trained for aligned CoT and not CoT (as with standard Transformers). Have you investigated this further or does there exist literature on this? I believe this could be an interesting research direction for scaling RNNs & SSMs.
2. The analysis of the sharing factor suggests that Transformers learn isolated solutions ("memorization"). How would this relate to e.g. the universal Transformer that explicitly ties weigths across layers? Do you think this would also create an "induction bias" similar to recurrent models?

**Limitations:**

The paper includes a dedicated section on the societal implications. However, a dedicated paragraph in the conclusion on the technical limitations of the study would strengthen the paper.

**Strengths And Weaknesses:**

**Strengths:**
- The paper is well written and easy to follow. It rigorously states its methodology and presents clean empircial results.
- The study setup is clean and investigates an important problem in todays deep sequence models.
- The results are intriguing albeit on synthetic and toy tasks.
- The results on CoT and align CoT are practically relevant and have the potential to spark new research.
- The paper is well-situated in the literature, albeit very focused and narrow.

**Weaknesses:**
- The main weakness of the paper is its focus on synthetic and toy tasks. I will concede that focusing on synthetic tasks is often the only viable way to do these broad empirical studies. However, scaling up and showing results for the (two) main finding(s) would strengthen the paper.
- My main reservation in the methodology is with respect to the fixed optimization budget (250k optimization steps). Since this does not scale with the train set size $N$, the models either overfit or underfit and the resulting minimum sample size $N^*$ is biased. The optimization budget should either scale with $N$ or use a validation hold-out set during training.


**Minor Comments:**
- In the methodology section of the main text, the paper reports the interval $n \in [2, L]$, but in the appendix this is $n \in [1, L]$.
- The paper could benefit from a broader relation to existing work, this would better contextualize the findings in the wider literature.

---

> ### Author Rebuttal · Authors · 2026-03-31
>
> We thank the reviewer for their time and thoughtful comments and respond below.
>
> > The main weakness of the paper is its focus on synthetic and toy tasks. I will concede that focusing on synthetic tasks is often the only viable way to do these broad empirical studies. However, scaling up and showing results for the (two) main finding(s) would strengthen the paper.
>
> We appreciate the reviewer’s comment and agree that understanding state tracking behavior on real-world tasks would add further value. That being said, we would like to highlight additional benefits of using synthetic tasks, beyond allowing for performing the large-scale evaluation as done in this work:
>
> First, the tasks used in the paper are not intended as merely toy benchmarks, rather they represent core abstractions of sequential dependency in temporal data. Specifically, certain essential substructures within real-world state-tracking tasks are equivalent to performing algebraic group computations [1]. For example, prior work [2] shows that $S_5$ (used in the weight-sharing experiments) can be reduced to chess state tracking, and highlights strong connections to code execution and entity tracking.
>
> Second, synthetic group problems offer significant advantages for a controlled study. They are precisely defined and algebraically analyzable. More importantly, they naturally allow precise control over state size ($m$) and number of state updates ($n$), enabling clear measurement of how data efficiency scales with these factors. These tasks also allowed us to evaluate state‑tracking in isolation without confounding factors such as language understanding, dataset bias, memorization, or heuristic exploitation.
>
> Moreover, the two tasks used in the paper represent canonical group structures: modular addition (cyclic group $Z_m$) is the canonical commutative task, since every finite abelian group is isomorphic to a direct product of cyclic groups. Similarly, permutation composition (symmetric group $S_m$) represents the canonical non‑commutative group structure, as any finite group  is isomorphic to a subgroup of a symmetric group.
>
> [1] Liu, B. et al. Transformers learn shortcuts to automata. ICLR, 2023
>
> [2] Merrill, W. et al. The Illusion of State in State‑Space Models. ICML, 2024.
>
> > My main reservation in the methodology is with respect to the fixed optimization budget (250k optimization steps) ... The optimization budget should either scale with $N$ or use a validation hold-out set during training.
>
> We thank the reviewer for pointing out this misunderstanding. We do use early stopping based on a hold‑out validation set, so $250$K refers to the _maximum_ number of optimization steps, regardless of the training set size. We acknowledge this was not clear in the manuscript, and we have now clarified it in the text.
>
> >In the methodology section of the main text, the paper reports the interval $n\in[2,L]$, but in the appendix this is $n\in[1, L]$.
>
> Fixed, thank you.
>
> >The paper could benefit from a broader relation to existing work...
>
> We have substantially revised the Related Work section accordingly to better situate our contributions in the literature.
>
> >The results regarding CoT and aligned CoT suggest that RNNs & SSMs at large scale should be trained for aligned CoT and not CoT. Have you investigated this further...
>
> While there is some evidence for this finding scattered in the literature (for example, [3] explain the challenges of CoT for RNNs through a recall bottleneck, and [4] proposes Aligned CoT for transformers) it seems to be not well-known, although it may be an important finding to take into consideration in future work on RNNs and SSMs.
>
> [3] Wen, et al., RNNs are not Transformers (Yet): The Key Bottleneck on In-Context Retrieval. ICLR 2025.
>
> [4] Zhang et al., Grounding code understanding in step-by-step execution, 2025.
>
> >The analysis of the sharing factor suggests that Transformers learn isolated solutions ("memorization"). How would this relate to e.g. the universal Transformer that explicitly ties weigths across layers? Do you think this would also create an "induction bias" similar to recurrent models?
>
> This is an interesting question. We would argue that models like the universal transformer (and other looped transformer variants) do not create an induction bias as this relies both on recurrence and temporal locality. These models add recurrence (through layers and not time) but are processing the input globally. However, we agree that the set of capabilities of these models as compared to standard recurrent networks is an important open question.
>
> > ... dedicated paragraph in the conclusion on the technical limitations ...
>
> We agree and will include a more in-depth discussion of limitations in the final version, where we will mention limitations with respect to the number of tasks, different models, and gaining a deeper empirical understanding of how the findings translate to real-world tasks.

---

> > ### Author Rebuttal · Reviewer_HvxJ · 2026-04-04
> >
> > Thank you for the rebuttal.
> >
> > The authors addressed almost all of my concerns and questions. The remaining one is simply to scale up one of the experiments. If the authors are able to include this in the final submission, I'm happy to raise my score. For now I will maintain my score.

---

> > > ### Author Response · Authors · 2026-04-07
> > >
> > > Thank you for your engagement and positive assessment. We’re glad almost all concerns have been addressed. Following your suggestion, we plan to include results on chess state tracking as a real-world task in the final version, complementing the synthetic group tasks and addressing the remaining concern about scale. We appreciate your consideration.

---

### Official Review · Reviewer_BaLn · 2026-03-13

**Soundness:** 4
**Presentation:** 2
**Significance:** 3
**Originality:** 3
**Overall Recommendation:** 5
**Confidence:** 4

**Summary:**

In its core, this work analyses the number of samples required for three representative model classes (Transformer, Nonlinear RNN and Linear Selective SSM) using three supervision techniques (Final state supervision, intermediate state supervision after each input, intermediate state supervision after the whole input) to learn to accurately predict the state of a finite-state automaton after being fed a sequence of inputs.

**Compliance With Llm Reviewing Policy:**

Affirmed.

**Final Justification:**

The authors have appropriately addressed my concerns during the rebuttal period, after which I have decided to raise my score from "weak accept" to "accept".

**Key Questions For Authors:**

I do not have any clarifying questions. Everything addressable is listed as a weakness above.

**Limitations:**

yes

**Strengths And Weaknesses:**

**Strengths**
- This is an extensive study of in-domain performance, or rather sample efficiency, of various models under various supervision regimes and other experimental setups, on the task of finite-state automaton emulation. It is the largest and most extensive study of its kind thus far, to the best of my knowledge, and it has interesting insights.
- The analysis is solid and delivers very clear messages, which are also very easy to read out from the excellently designed Page 6.
- The storyline is clear.

**Weaknesses**
- The SSM model selection is questionable; such a bilinear RNN does indeed perform well on state tracking, but seems to me to be of little utility for more general tasks where input accumulation is critical. Adding a Mamba/DeltaNet/any one of the growing family of SSMs alongside the existing results would make this study much more impactful.
- The considered task induces a very clear picture in terms of the sample efficiency of the models, but these are still variants of a very simple base task. If resources allow it, an analysis like this on a different task with controllable difficulty would be a valuable addition to the paper.

**Presentation Weaknesses/Suggestions/Questions:**
- In the introduction, when discussing the investigation of state tracking with Transformers, very little prior work is provided; arguably, Anil et al., and Dziri et al. are not the central citations, but rather maybe [1,2,3,4,5,6,7,8].
- The reasoning in the 1st sentence of the 1st paragraph of page 2 (continuing from page 1) is unclear to me. It reads a bit contradictory to me, here's how I parsed it: "The transformer's output depends on all past inputs *and outputs*, and the RNN's only depends on the hidden state, therefore the Transformer's computation depends on the global history, but the RNN's does not". I understand what you meant, but it reads differently than what was intended. The sentence afterwards is valid.
- The induction bias is intuitively clear but it is not very well defined; maybe "Markovian condition" would be better nomenclature? You might want to check for counterexamples, which [8] might be.
- I find the seemingly quadratic fit of the lines in Figure 2 very confusing. The solid orange line goes exactly through one orange square and two orange circles. The growth seems to be overestimated without knowledge of further datapoints which can be found in Table 1 but are not visible in the figure. I think a denser figure with more datapoints would be a better fit here.


I hope the years are correct in the below list:
[1] - M. Hahn, 2019, "Theoretical Limitations of Self-Attention in Neural Sequence Models"
[2] - Bhattamishra et al., 2020, "On the Ability and Limitations of Transformers to Recognize Formal Languages"
[3] - Deletang et al., 2023, "Neural Networks and the Chomsky Hierarchy"
[4] - Merrill et al., 2024, "The Expressive Power of Transformers with Chain of Thought"
[5] - Liu et al., 2023, "Transformers Learn Shortcuts to Automata"
[6] - Strobl et al., 2024, "What Formal Languages can Transformers Express? A Survey"
[7] - Chi et al., 2023, "Transformer Working Memory Enables Regular Language Reasoning And Natural Language Length Extrapolation"

---

> ### Author Rebuttal · Authors · 2026-03-31
>
> We thank the reviewer for their time and thoughtful feedback. We respond to the comments below.
>
> > The SSM model selection
>
> Thank you for your suggestion. We included the bilinear model in our study, because of its strong state tracking capability: prior work has shown that this type of linear RNN can simulate any FSA within a single layer. However, as you noted, this model is not equipped with explicit memory mechanisms. Its purpose in our study is therefore to represent one end of the spectrum, pure state tracking, while Transformers represent the other end, with strong in‑context memory but weaker explicit state tracking. We will include a brief discussion on this in the revised version of the text.
>
> We agree that including more widely used SSM variants would improve the study. Following your suggestion, we added Mamba to the weight‑sharing experiments (Section 4). We include only the sharing factors here and refer to [this link](https://imgur.com/rTpjjya) for the updated figure:
>
> ||Mamba|Transformer|LSTM|Dense-SSM|
> |-|-|-|-|-|
> |OutcomeSupervision|3.62|0.82|21.09|35.13|
> |CoT|0.30|0.28|1.03|1.11|
> |AlignedCoT|2.49|1.12|4.10|9.83|
>
> Mamba shows a slight improvement over transformers in terms of sharing factor, however, this behavior is consistent with prior observations that Mamba exhibits state tracking limitations similar to transformers [1]. We will update the manuscript with this result and, if time permits, also include a DeltaNet variant for completeness.
>
> [1] Grazzi, et al. Unlocking State-Tracking in Linear RNNs Through Negative Eigenvalues. ICLR 2025.
>
> >The considered task induces a very clear picture ... but still variants of a very simple base task
>
> We agree including other tasks would be valuable. Besides the already substantial computational challenges associated with this study, we would like to point out the following:
>
> 1. Besides modular addition, we do use the permutation‑composition task (symmetric group $S_5$) for the weight‑sharing experiments, and reported similar results to the case of modular addition.
>
> 2. Despite its simplicity, modular addition (cyclic group $\mathbb{Z}_m$) is the canonical commutative state‑tracking task, since every finite abelian group is isomorphic to a direct product of cyclic groups. A useful property of this task is that it allows easy control over the state size (via the modulus $m$) without fundamentally increasing the difficulty of the problem.
>
> >  prior work
>
> We thank the reviewer for highlighting the references. We have substantially revised the Related Work section and included the suggested papers in the discussion for the final version.
>
> >The reasoning in the 1st sentence of page 2 is unclear
>
> Thank you for pointing this out.
> We have reformulated the statement to make it clear that it refers to an inductive bias not a deficiency of the RNN:
>
> A key difference between transformer-based and recurrent models is that, at every timestep, the former compute outputs by applying a function that depends on all inputs and outputs generated previously (the context window), making it possible, in principle, to re-calculate the required state from the past information globally at each time step.
> Recurrent networks, on the other hand, condition each output on a hidden state that is incrementally updated from the input and carried forward over time, which prevents recomputing the state from the full history at each timestep.
>
> We hope this clarifies our intent and would appreciate knowing whether this addresses your concern.
>
> > The induction bias is intuitively clear but it is not very well defined...
>
> We added the following definition of induction bias in the introduction:
>
> Formally, the presence of the induction bias in a model means
> that the joint distribution over tokens, conditioned on
> the most recent hidden state, factorizes, such that
> $p(x_{t+1}|x_1, \ldots, x_{t},h_{t})=p(x_{t+1}| h_{t})$,
> where $x_t$ is the $t$-th token and $h_{t}$
> the hidden state in timestep $t$, representing a minimal
> sufficient statistic for determining $x_{t+1}$.
>
> (It seems the reviewer may have not provided a reference [8], as only seven references were listed in the comment.)
>
> > quadratic fit of Figure 2
>
> Thank you for pointing this out. In Figure 2, the y-axis was capped at 10K as the y-range of the samples needed for models with and without their preferred supervision methods are drastically different. We emphasize that the regression lines were fit using all data points from Table 1, but the limited axis range made this unclear.
>
> We have revised the figure to show the full y-axis range and to plot all available data points. We also updated the trend lines across the paper to use a poly-logarithmic fit, which avoids negative extrapolation and improves the visual clarity of the plots.

---

> > ### Author Rebuttal · Reviewer_BaLn · 2026-04-04
> >
> > I thank the authors for addressing my concerns and clarifying certain aspects of their work in their rebuttal. I am happy to raise my score to an accept.

---

### Decision · Program_Chairs · 2026-04-30

**Decision:**

Accept (regular)

**Comment:**

The paper presents an empirical study showing that transformers have much worse in-distribution data efficiency than RNNs/SSMs on state-tracking tasks, tracing this to negligible cross-length weight sharing.

In other words, transformers learn length-specific solutions while recurrent models share mechanisms across lengths. The CoT vs  aligned-CoT findings are also interesting.

Reviewers generally appreciated the results. Reviewers also raised concerns about the reliance on synthetic tasks and limited architectural coverage. The authors addressed these with new Mamba results in the weight-sharing experiments. I encourage the authors to incorporate these into the revision.